# TopoTune: A Framework for Generalized Combinatorial Complex Neural Networks

**Mathilde Papillon** [1]  **Guillermo Bernárdez** [1]  **Claudio Battiloro** [*2]  **Nina Miolane** [*1]

## Abstract

Graph Neural Networks (GNNs) effectively learn from relational data by leveraging graph symmetries. However, many real-world systems—such as biological or social networks—feature multiway interactions that GNNs fail to capture. Topological Deep Learning (TDL) addresses this by modeling and leveraging higher-order structures, with Combinatorial Complex Neural Networks (CCNNs) offering a general and expressive approach that has been shown to outperform GNNs. However, TDL lacks the principled and standardized frameworks that underpin GNN development, restricting its accessibility and applicability. To address this issue, we introduce Generalized CCNNs (GCCNs), a simple yet powerful family of TDL models that can be used to systematically transform any (graph) neural network into its TDL counterpart. We prove that GCCNs generalize and subsume CCNNs, while extensive experiments on a diverse class of GCCNs show that these architectures consistently match or outperform CCNNs, often with less model complexity. In an effort to accelerate and democratize TDL, we introduce TopoTune, a lightweight software for defining, building, and training GCCNs with unprecedented flexibility and ease.

## 1. Introduction

Graph Neural Networks (GNNs) (Scarselli et al., 2008; Corso et al., 2024) have demonstrated remarkable performance in several relational learning tasks by incorporating prior knowledge through graph structures (Kipf & Welling, 2017; Zhang & Chen, 2018). However, constrained by the pairwise nature of graphs, GNNs are limited in their ability to capture and model higher-order interactions—crucial in complex systems like particle physics, social interactions, or biological networks (Lambiotte et al., 2019). *Topological Deep Learning* (TDL) (Bodnar, 2023; Battiloro, 2024) precisely emerged as a framework that naturally encompasses multi-way relationships, leveraging beyond-graph combinatorial topological domains such as simplicial and cell complexes, or hypergraphs (Papillon et al., 2023).[1]

In this context, Hajij et al. (2023; 2024a) have recently introduced *combinatorial complexes*, fairly general objects that are able to model *arbitrary* higher-order interactions along with a *hierarchical* organization among them–hence generalizing (for learning purposes) most of the combinatorial topological domains within TDL, including graphs. The elements of a combinatorial complex are *cells*, being nodes or groups of nodes, which are categorized by *ranks*. The simplest cell, a single node, has rank zero. Cells of higher ranks define relationships between nodes: rank one cells are edges, rank two cells are faces, and so on. Hajij et al. (2023) also proposes *Combinatorial Complex Neural Networks* (CCNNs), deep learning architectures that leverage the versatility of combinatorial complexes to naturally model higher-order interactions. For instance, consider the task of predicting the solubility of a molecule from its structure. GNNs model molecules as graphs, thus considering atoms (nodes) and bonds (edges) (Gilmer et al., 2017). By contrast, CCNNs model molecules as combinatorial complexes, hence considering atoms (nodes, i.e., cells of rank zero), bonds (edges, i.e., cells of rank one), and also important higher-order structures such as rings or functional groups (i.e., cells of rank two) (Battiloro et al., 2025).

**TDL Research Trend.** To date, research in TDL has largely progressed by taking existing GNNs architectures (convolutional, attentional, message-passing, etc.) and generalizing them one-by-one to a specific TDL counterpart, whether that be on hypergraphs (Feng et al., 2019; Chen et al., 2020a; Yadati, 2020), on simplicial complexes (Roddenberry et al., 2021; Yang & Isufi, 2023; Ebli et al., 2020; Giusti et al., 2022a; Battiloro et al., 2024; Bodnar et al.,

---
[*]Equal contribution [1]University California Santa Barbara, USA [2]Harvard University, USA. Correspondence to: Mathilde Papillon <papillon@ucsb.edu>.

*Proceedings of the 42nd International Conference on Machine Learning*, Vancouver, Canada. PMLR 267, 2025. Copyright 2025 by the author(s).

---
[1]Simplicial and cell complexes model *specific* higher-order interactions organized *hierarchically*, while hypergraphs model *arbitrary* higher-order interactions but *without any hierarchy*.

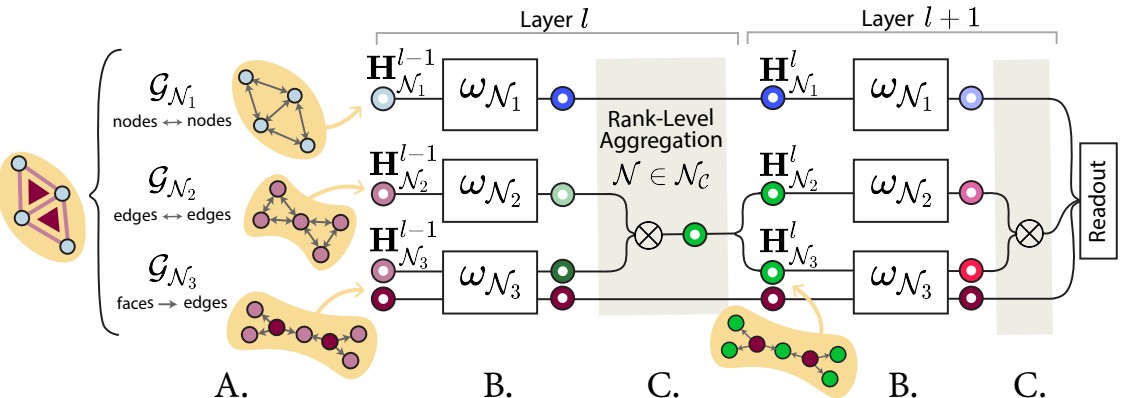

*Figure 1.* **Generalized Combinatorial Complex Network (GCCN).** The input complex $\mathcal{C}$ has neighborhoods $\mathcal{N}_\mathcal{C} = \{\mathcal{N}_1, \mathcal{N}_2, \mathcal{N}_3\}$. **A.** The complex is expanded into three augmented Hasse graphs $\mathcal{G}_{\mathcal{N}_i}$, $i = \{1, 2, 3\}$, each with features $H_{\mathcal{N}_i}$ represented as a colored disc. **B.** A GCCN layer dedicates one base architecture $\omega_{\mathcal{N}_i}$ (GNN, Transformer, MLP, etc.) to each neighborhood. **C.** The output of all the architectures $\omega_{\mathcal{N}_i}$ is aggregated rank-wise, then updated. In this example, only the complex's edge features (originally pink) are aggregated across multiple neighborhoods ($\mathcal{N}_2$ and $\mathcal{N}_3$).

2021b; Maggs et al., 2024; Lecha et al., 2025), on cell complexes (Hajij et al., 2020; Giusti et al., 2022b; Bodnar et al., 2021a), or on combinatorial complexes (Battiloro et al., 2025; Eitan et al., 2024). Although overall valuable and insightful, such a fragmented research trend is slowing the development of standardized methodologies and software for TDL, as well as limiting the analysis of its cost-benefits trade-offs (Papamarkou et al., 2024). We argue these two challenges are hindering the use and application of TDL beyond the community of experts. This is particularly relevant as practitioners are beginning to turn to TDL for tackling application-specific scenarios, such as computer network modelling (Bernárdez et al., 2025).

**Current Efforts and Gaps for TDL Standardization.**
TopoX (Hajij et al., 2024b) and TopoBench(Telyatnikov et al., 2024) have become the reference Python libraries for *developing* and *benchmarking* TDL models, respectively. However, despite their potential in defining and implementing novel standardized methodologies in the field, the current focus of these packages is on replicating and analyzing existing message-passing CCNNs. Works like Jogl et al. (2022b;a) have instead focused on making TDL accessible by porting models to the graph domain. They do so via principled transformations from combinatorial topological domains to graphs. However, although these architectures are as expressive as their TDL counterparts (using the Weisfeiler-Lehman criterion (Xu et al., 2019b)), they are neither formally equivalent to nor a generalization of their TDL counterparts. Due to collapse of rank information during the graph expansion, the GNNs on the resulting graph do not preserve the same topological information.

**Contributions.** This works seeks to accelerate TDL research and increase its accessibility and standardization for outside practitioners. To that end, we introduce a novel joint methodological and software framework that easily enables the development of new TDL architectures in a principled way—overcoming the limitations of existing works. We outline our main contributions and specify which of the field's open problems (OPs), as defined in Papamarkou et al. (2024), they help answer:

- **Systematic Generalization.** We propose the first method to systematically generalize *any neural network* to its topological counterpart with minimal adaptation. Specifically, we define a novel expansion mechanism that transforms a combinatorial complex into a collection of graphs, enabling the training of TDL models as an ensemble of synchronized models. To our knowledge, this is the first method designed to accommodate many topological domains. (OPs 6, 11: foundational, cross-domain TDL.)

- **General Architectures.** Our method induces a novel wide class of TDL architectures, *Generalized Combinatorial Complex Networks* (GCCNs), portrayed in Fig. 1. GCCNs *(i)* formally generalize CCNNs, *(ii)* are cell permutation equivariant, and *(iii)* are as expressive as CCNNs. (OP 9: consolidating TDL advantages in a unified theory.)

- **Implementation.** We provide TopoTune, a lightweight PyTorch module for developing GCCNs fully integrated into TopoBench (Telyatnikov et al., 2024). (OP 4: need for software). Using TopoTune, practitioners can, for the first time, easily define and iterate upon TDL models, making TDL a much more practical tool for real-world datasets (OP 1: need for accessible TDL).

- **Benchmarking.** Using TopoTune, we create a broad class of GCCNs using four base GNNs and one base Trans-

former over two combinatorial topological spaces (simplicial and cell complexes). Unlike prior works that compare models under heterogeneous conditions, our systematic benchmarking provides a controlled evaluation of GCCNs across diverse architectures, datasets, and topological domains. A wide range of experiments on graph-level and node-level benchmark datasets shows GCCNs generally outperform existing CCNNs, often with smaller model sizes. Some of these results are obtained with GCCNs that cannot be reduced to standard CCNNs, further underlining our methodological contribution. A guide to the code is available at geometric-intelligence.github.io/topotune. (OP 3: need for standardized benchmarking.)

**Outline.** Section 2 provides necessary background. Section 3 motivates and positions our work in the current TDL literature. Section 4 introduces and discusses GCCNs. Section 5 describes TopoTune. Finally, Section 6 showcases extensive numerical experiments and comparisons.

## 2. Background

To properly contextualize our work, we revisit the fundamentals of combinatorial complexes and CCNNs—closely following the works of Hajij et al. (2023) and Battiloro et al. (2025)—as well as the notion of augmented Hasse graphs. Appendix A briefly introduces all topological domains used in TDL, such as simplicial and cell complexes.

**Combinatorial Complex.** A *combinatorial complex* is a triple $(\mathcal{V}, \mathcal{C}, \mathrm{rk})$ consisting of a set $\mathcal{V}$, a subset $\mathcal{C}$ of the powerset $\mathcal{P}(\mathcal{V}) \backslash \{\emptyset\}$, and a rank function $\mathrm{rk} : \mathcal{C} \to \mathbb{Z}_{\geq 0}$ with the following properties:

1. for all $v \in \mathcal{V}$, $\{v\} \in \mathcal{C}$ and $\mathrm{rk}(\{v\}) = 0$;

2. the function $\mathrm{rk}$ is order-preserving, i.e., if $\sigma, \tau \in \mathcal{C}$ satisfy $\sigma \subseteq \tau$, then $\mathrm{rk}(\sigma) \leq \mathrm{rk}(\tau)$.

The elements of $\mathcal{V}$ are the nodes, while the elements of $\mathcal{C}$ are called cells (i.e., group of nodes). The rank of a cell $\sigma \in \mathcal{C}$ is $k := \mathrm{rk}(\sigma)$, and we call it a $k$-cell. $\mathcal{C}$ simplifies notation for $(\mathcal{V}, \mathcal{C}, \mathrm{rk})$, and its dimension is defined as the maximal rank among its cell: $\dim(\mathcal{C}) := \max_{\sigma \in \mathcal{C}} \mathrm{rk}(\sigma)$.

**Neighborhoods.** Combinatorial complexes can be equipped with a notion of neighborhood among cells. In particular, a neighborhood $\mathcal{N} : \mathcal{C} \to \mathcal{P}(\mathcal{C})$ on a combinatorial complex $\mathcal{C}$ is a function that assigns to each cell $\sigma$ in $\mathcal{C}$ a collection of "neighbor cells" $\mathcal{N}(\sigma) \subset \mathcal{C} \cup \emptyset$. Examples of neighborhood functions are *adjacencies*, connecting cells with the same rank, and *incidences*, connecting cells with different consecutive ranks. Usually, up/down incidences

$\mathcal{N}_{I,\uparrow}$ and $\mathcal{N}_{I,\downarrow}$ are defined as

$$\begin{aligned}
\mathcal{N}_{I,\uparrow}(\sigma) &= \big\{ \tau \in \mathcal{C} \,\big|\, \mathrm{rk}(\tau) = \mathrm{rk}(\sigma) + 1, \, \sigma \subset \tau \big\}, \\
\mathcal{N}_{I,\downarrow}(\sigma) &= \big\{ \tau \in \mathcal{C} \,\big|\, \mathrm{rk}(\tau) = \mathrm{rk}(\sigma) - 1, \, \tau \subset \sigma \big\}.
\end{aligned} \quad (1)$$

Therefore, a $k + 1$-cell $\tau$ is a neighbor of a $k$-cell $\sigma$ w.r.t. to $\mathcal{N}_{I,\uparrow}$ if $\sigma$ is contained in $\tau$; analogously, a $k - 1$-cell $\tau$ is a neighbor of a $k$-cell $\sigma$ w.r.t. to $\mathcal{N}_{I,\downarrow}$ if $\tau$ is contained in $\sigma$. These incidences induce up/down adjacencies $\mathcal{N}_{A,\uparrow}$ and $\mathcal{N}_{A,\downarrow}$ as

$$\begin{aligned}
\mathcal{N}_{A,\uparrow}(\sigma) &= \big\{ \tau \in \mathcal{C} \,\big|\, \mathrm{rk}(\tau) = \mathrm{rk}(\sigma), && (2) \\
&\quad \exists \delta \in \mathcal{C} : \mathrm{rk}(\delta) = \mathrm{rk}(\sigma) + 1, \, \tau \subset \delta, \, \sigma \subset \delta \big\}, \\
\mathcal{N}_{A,\downarrow}(\sigma) &= \big\{ \tau \in \mathcal{C} \,\big|\, \mathrm{rk}(\tau) = \mathrm{rk}(\sigma), && (3) \\
&\quad \exists \delta \in \mathcal{C} : \mathrm{rk}(\delta) = \mathrm{rk}(\sigma) - 1, \, \delta \subset \tau, \, \delta \subset \sigma \big\}.
\end{aligned}$$

Therefore, a $k$-cell $\tau$ is a neighbor of a $k$-cell $\sigma$ w.r.t. to $\mathcal{N}_{A,\uparrow}$ if they are both contained in a $k + 1$-cell $\delta$; analogously, a $k$-cell $\tau$ is a neighbor of a $k$-cell $\sigma$ w.r.t. to $\mathcal{N}_{A,\downarrow}$ if they both contain a $k - 1$-cell $\delta$. Other neighborhood functions can be defined for specific applications (Battiloro et al., 2025).

**Combinatorial Complex Message-Passing Neural Networks.** Let $\mathcal{C}$ be a combinatorial complex, and $\mathcal{N}_{\mathcal{C}}$ a collection of neighborhood functions. The $l$-th layer of a CCNN updates the embedding $\mathbf{h}_\sigma^l \in \mathbb{R}^{F^l}$ of cell $\sigma$ as

$$\mathbf{h}_\sigma^{l+1} = \phi \left( \mathbf{h}_\sigma^l, \bigotimes_{\mathcal{N} \in \mathcal{N}_{\mathcal{C}}} \bigoplus_{\tau \in \mathcal{N}(\sigma)} \psi_{\mathcal{N}, \mathrm{rk}(\sigma)} \left( \mathbf{h}_\sigma^l, \mathbf{h}_\tau^l \right) \right) \in \mathbb{R}^{F^{l+1}},$$

(4)

where $\mathbf{h}_\sigma^0 := \mathbf{h}_\sigma$ are the initial features, $\bigoplus$ is an intra-neighborhood aggregator, $\bigotimes$ is an inter-neighborhood aggregator. The functions $\psi_{\mathcal{N}, \mathrm{rk}(\cdot)} : \mathbb{R}^{F^l} \to \mathbb{R}^{F^{l+1}}$ and the update function $\phi$ are learnable functions, which are typically homogeneous across all neighborhoods and ranks. In other words, the embedding of a cell is updated in a learnable fashion by first aggregating messages with neighboring cells per each neighborhood, and then by further aggregating across neighborhoods. We remark that by this definition, all CCNNs are message-passing architectures. Moreover, they can only leverage neighborhood functions that consider all ranks in the complex.

**Augmented Hasse Graphs.** In TDL, a Hasse graph is a graph expansion of a combinatorial complex. Specifically, it represents the incidence structure $\mathcal{N}_{I,\downarrow}$ by representing each cell (node, edge, face) as a node and drawing edges between cells that are incident to each other. Going beyond just considering $\mathcal{N}_{I,\downarrow}$, given a collection of *multiple* neighborhood functions, a combinatorial complex $\mathcal{C}$ can be expanded into a unique graph representation. We refer to this representation as an augmented Hasse graph (Fig. 2) (Hajij et al., 2023). Formally, let $\mathcal{N}_{\mathcal{C}}$ be a collection of neighborhood

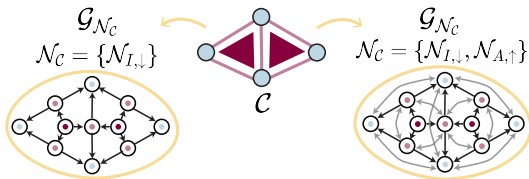

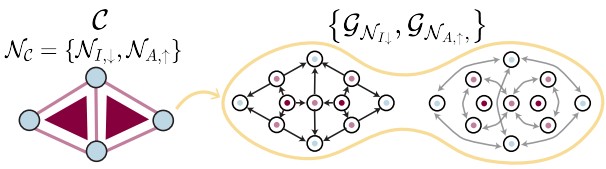

*Figure 2.* **Augmented Hasse graphs.** Expansions of a combinatorial complex $\mathcal{C}$ (middle) into two augmented Hasse graphs: (left) the Hasse graph induced by $\mathcal{N}_{\mathcal{C}} = \{\mathcal{N}_{I,\downarrow}\}$; (right) the augmented Hasse graph induced by $\mathcal{N}_{\mathcal{C}} = \{\mathcal{N}_{I,\downarrow}, \mathcal{N}_{A,\uparrow}\}$. Information on cell rank is discarded (we retain rank color for illustrative purposes).

*Figure 3.* **Ensemble of strictly augmented Hasse Graphs.** Given a complex $\mathcal{C}$ with neighborhood structure including both incidence and upper adjacency (left), this graph expansion (right) produces one augmented Hasse graph for *each* neighborhood.

functions on $\mathcal{C}$: the augmented Hasse graph $\mathcal{G}_{\mathcal{N}_{\mathcal{C}}}$ of $\mathcal{C}$ induced by $\mathcal{N}_{\mathcal{C}}$ is a directed graph $\mathcal{G}_{\mathcal{N}_{\mathcal{C}}} = (\mathcal{C}, \mathcal{E}_{\mathcal{N}_{\mathcal{C}}})$ with cells represented as nodes, and edges given by

$$\mathcal{E}_{\mathcal{N}_{\mathcal{C}}} = \{(\tau, \sigma) | \sigma, \tau \in \mathcal{C}, \exists \mathcal{N} \in \mathcal{N}_{\mathcal{C}} : \tau \in \mathcal{N}(\sigma)\}. \quad (5)$$

The augmented Hasse graph of $\mathcal{C}$ is thus obtained by considering the cells as nodes, and inserting directed edges among them if the cells are neighbors in $\mathcal{C}$. Notably, such a representation of a combinatorial complex discards all information about cell rank.

## 3. Motivation and Related Works

As outlined in the introduction, TDL lacks a comprehensive framework for easily creating and experimenting with novel topological architectures—unlike the more established GNN field. This section outlines some previous works that have laid important groundwork in addressing this challenge.

**Formalizing CCNNs on graphs.** The position paper (Veličković, 2022) proposed that any function over a higher-order domain can be computed via message passing over a transformed graph, but without specifying how to design GNNs that reproduce CCNNs. Hajij et al. (2023) showed that, given a combinatorial complex $\mathcal{C}$ and a collection of neighborhoods $\mathcal{N}_{\mathcal{C}}$, a message-passing GNN that runs over the augmented Hasse graph $\mathcal{G}_{\mathcal{N}_{\mathcal{C}}}$ is equivalent to a specific CCNN as in (4) running over $\mathcal{C}$ using: i) $\mathcal{N}_{\mathcal{C}}$ as collection of neighborhoods; ii) same intra- and inter-aggregations, i.e., $\bigoplus = \bigotimes$; and iii) no rank- and neighborhood-dependent message functions, i.e., $\psi_{\mathcal{N}, \text{rk}(\cdot)} = \psi \; \forall \mathcal{N} \in \mathcal{N}_{\mathcal{C}}$.

**Retaining expressivity, but not topological symmetry.** Jogl et al. (2022a;b) demonstrate that GNNs on augmented Hasse graphs $\mathcal{G}_{\mathcal{N}_{\mathcal{C}}}$ are as expressive as CCNNs on $\mathcal{C}$ (using the WL criterion), suggesting that some CCNNs can be simulated with standard graph libraries. [2] However, as the authors state, such GNNs do not structurally distinguish between cells of different ranks or neighborhoods, collapsing

topological relationships. For instance, in a molecule (cellular complex), two bonds (edges) may simultaneously share multiple neighborhoods: lower-adjacent through a shared atom (node) and upper-adjacent through a shared ring (face). A GNN on $\mathcal{G}_{\mathcal{N}_{\mathcal{C}}}$ collapses these distinctions, applying the same weights to all connections and losing the structural symmetries encoded in the domain. While this may suffice for preserving expressivity, it is inherently a very different computation than that of TDL models.

**The Particular Case of Hypergraphs.** Hypergraph neural networks have long relied on graph expansions (Telyatnikov et al., 2023), which has allowed the field to leverage advances in the graph domain and, by extension, a much wider breadth of models (Antelmi et al., 2023; Papillon et al., 2023). Most hypergraph models are expanded into graphs using the star (Zhou et al., 2006; Solé et al., 1996), the clique (Bolla, 1993; Rodríguez, 2002; Gibson et al., 2000), or the line expansion (Bandyopadhyay et al., 2020). As noted by Agarwal et al. (2006), many hypergraph learning algorithms leverage graph expansions.

The success story of hypergraph neural networks motivates further research on new graph-based expansions of CCNNs. These expansions could, at the same time, subsume current CCNNs *and* exploit progress in the GNN field. Therefore, returning to our core goal of accelerating and democratizing TDL while preserving its theoretical properties, we propose a two-part approach: a **novel graph-based methodology** able to generate general architectures (Section 4), and a **lightweight software framework** to easily and widely implement it (Section 5).

## 4. Generalized Combinatorial Complex Neural Networks

We propose Generalized Combinatorial Complex Neural Networks (GCCNs), a novel broad class of TDL architectures. GCCNs overcome the limitations of previous graph-based TDL architectures by leveraging the notions of *strictly augmented Hasse graphs* and *per-rank neighborhoods*.

---

[2]The same authors generalize these ideas to non-standard message-passing GNNs (Jogl et al., 2024).

**Ensemble of Strictly Augmented Hasse Graphs.** This graph expansion method (see Fig. 3) extends from the established definition of an augmented Hasse graph (see Fig. 2). Specifically, given a combinatorial complex $\mathcal{C}$ and a collection of neighborhood functions $\mathcal{N}_{\mathcal{C}}$, we expand it into $|\mathcal{N}_{\mathcal{C}}|$ graphs, each of them representing a neighborhood $\mathcal{N} \in \mathcal{N}_{\mathcal{C}}$. In particular, the *strictly augmented Hasse graph* $\mathcal{G}_{\mathcal{N}} = (\mathcal{C}_{\mathcal{N}}, \mathcal{E}_{\mathcal{N}})$ of a neighborhood $\mathcal{N} \in \mathcal{N}_{\mathcal{C}}$ is a directed graph whose nodes $\mathcal{C}_{\mathcal{N}}$ and edges $\mathcal{E}_{\mathcal{N}}$ are given by:

$$\mathcal{C}_{\mathcal{N}} = \{\sigma \in \mathcal{C} \,|\, \mathcal{N}(\sigma) \neq \emptyset\},\ \mathcal{E}_{\mathcal{N}} = \{(\tau, \sigma)\,|\,\tau \in \mathcal{N}(\sigma)\}. \tag{6}$$

Following the same arguments from (Hajij et al., 2023), a GNN over the strictly augmented Hasse graph $\mathcal{G}_{\mathcal{N}}$ induced by $\mathcal{N}$ is equivalent to a CCNN running over $\mathcal{C}$ and using $\mathcal{N}_{\mathcal{C}} = \{\mathcal{N}\}$ up to the (self-)update of the cells in $\mathcal{C}/\mathcal{C}_{\mathcal{N}}$.

**Per-rank Neighborhoods.** The standard definition of adjacencies and incidences given in Section 2 implies that they are applied to each cell regardless of its rank. For instance, consider a combinatorial complex of dimension two with nodes (0-cells), edges (1-cells), and faces (2-cells). Employing the down incidence $\mathcal{N}_{I,\downarrow}$ as in (1) means the edges must exchange messages with their endpoint nodes, and faces must exchange messages with the edges on their sides. It is impossible for edges to communicate while faces do not.

This limitation increases the computational cost of standard CCNNs while not always increasing the learning performance, as experiments will show. For this reason, we introduce *per-rank neighborhoods*, examples of which are depicted in Fig. 4. Formally, a per-rank neighborhood function $\mathcal{N}^r$ maps a cell $\sigma$ to the empty set if $\sigma$ is a cell of rank $r$. For example, the up/down $r$-*incidences* $\mathcal{N}^r_{I,\uparrow}$ and $\mathcal{N}^r_{I,\downarrow}$ are defined as

$$\mathcal{N}^r_{I,\uparrow}(\sigma) = \begin{cases} \{\tau \in \mathcal{C} \,|\, \mathrm{rk}(\tau) = \mathrm{rk}(\sigma) + 1,\ \sigma \subset \tau\}, \\ \qquad\text{if } \mathrm{rk}(\sigma) = r, \\ \emptyset, \quad\text{otherwise}, \end{cases}$$

$$\mathcal{N}^r_{I,\downarrow}(\sigma) = \begin{cases} \{\tau \in \mathcal{C} \,|\, \mathrm{rk}(\tau) = \mathrm{rk}(\sigma) - 1,\ \sigma \subset \tau\}, \\ \qquad\text{if } \mathrm{rk}(\sigma) = r, \\ \emptyset, \quad\text{otherwise}. \end{cases} \tag{7}$$

and the up/down $r$-*adjacencies* $\mathcal{N}^r_{A,\uparrow}$ and $\mathcal{N}^r_{A,\downarrow}$ can be obtained analogously. So, it is now straightforward to model a setting in which employing only $\mathcal{N}^1_{I,\downarrow}$ (Fig. 4iii) allows edges to exchange messages with their bounding nodes but not triangles with their bounding edges.

**Generating Graph-based TDL Architectures.** We use these notions to define a novel graph-based methodology for generating principled TDL architectures. Given a combinatorial complex $\mathcal{C}$ and a set $\mathcal{N}_{\mathcal{C}}$ of neighborhoods, the

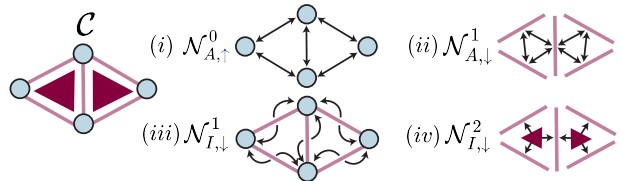

*Figure 4.* **Per-rank neighborhoods**. Given a complex $\mathcal{C}$ (left), we illustrate four examples of per-rank neighborhoods (right). In each case, they only include rank-specific cells.

method works as follows (see also Fig. 1): *(i)* $\mathcal{C}$ is expanded into an *ensemble* of strictly augmented Hasse graphs—one for each $\mathcal{N} \in \mathcal{N}_{\mathcal{C}}$. *(ii)* Each strictly augmented Hasse graph $\mathcal{G}_{\mathcal{N}}$ and the features of its cells are independently processed by a base model. *(iii)* An aggregation module $\bigotimes$ synchronizes the cell features across the different strictly augmented Hasse graphs (as the same cells can belong to multiple strictly augmented Hasse graphs).

This method enables an ensemble of synchronized models per layer— the $\omega_{\mathcal{N}}$s—each of them applied to a specific strictly augmented Hasse graph.[3] The rest of this section formalizes the architectures induced by this methodology.

**Generalized Combinatorial Complex Networks.** We formally introduce a broad class of novel TDL architectures called Generalized Combinatorial Complex Networks (GC-CNs), depicted in Fig. 1. Let $\mathcal{C}$ be a combinatorial complex containing $|\mathcal{C}|$ cells and $\mathcal{N}_{\mathcal{C}}$ a collection of neighborhoods on it. Assume an arbitrary labeling of the cells in the complex, and denote the $i$-th cell with $\sigma_i$. Denote by $\mathbf{H} \in \mathbb{R}^{|\mathcal{C}| \times F}$ the feature matrix collecting some embeddings of the cells on its rows, i.e., $[\mathbf{H}]_i = \mathbf{h}_{\sigma_i}$, and by $\mathbf{H}_{\mathcal{N}} \in \mathbb{R}^{|\mathcal{C}_{\mathcal{N}}| \times F}$ the submatrix containing just the embeddings of the cells belonging to the strictly augmented Hasse graph $\mathcal{G}_{\mathcal{N}}$ of $\mathcal{N}$. The $l$-th layer of a GCCN updates the embeddings of the cells $\mathbf{H}^l \in \mathbb{R}^{|\mathcal{C}| \times F^l}$ as

$$\mathbf{H}^{l+1} = \phi\left(\mathbf{H}^l, \bigotimes_{\mathcal{N} \in \mathcal{N}_{\mathcal{C}}} \omega_{\mathcal{N}}(\mathbf{H}^l_{\mathcal{N}}, \mathcal{G}_{\mathcal{N}})\right) \in \mathbb{R}^{|\mathcal{C}| \times F^{l+1}}, \tag{8}$$

where $\mathbf{H}^0$ collects the initial features, and the update function $\phi$ is a learnable row-wise update function, i.e., $[\phi(\mathbf{A}, \mathbf{B})]_i = \phi([\mathbf{A}]_i, [\mathbf{B}]_i)$. The neighborhood-dependent sub-module $\omega_{\mathcal{N}} : \mathbb{R}^{|\mathcal{C}_{\mathcal{N}}| \times F^l} \to \mathbb{R}^{|\mathcal{C}_{\mathcal{N}}| \times F^{l+1}}$, which we refer to as the *neighborhood message function*, is a learnable (matrix) function that takes as input the whole strictly augmented Hasse graph of the neighborhood, $\mathcal{G}_{\mathcal{N}}$ and the embeddings of the cells that are part of it, and gives as output a processed version of them. Finally, the inter-neighborhood

---

[3] Contrary to past CCNN simulation works that apply a model to the singular, whole augmented Hasse graph.

aggregation module $\bigotimes$ synchronizes the possibly multiple neighborhood messages arriving on a single cell across multiple strictly augmented Hasse graphs into a single message. GCCNs enjoy increased flexibility over CCNS (eq. 4) as their neighborhoods are allowed to be rank-dependent and their $\omega_\mathcal{N}$'s are not necessarily message-passing.

---

**Theoretical properties of GCCNs.**

1. **Generality.** GCCNs formally generalize CC-NNs.

   **Proposition 4.1.** *Let $\mathcal{C}$ be a combinatorial complex. Let $\mathcal{N}_\mathcal{C}$ be a collection of neighborhoods on $\mathcal{C}$. Then, there exists a GCCN that exactly reproduces the computation of a CCNN over $\mathcal{C}$ using $\mathcal{N}_\mathcal{C}$.*

   Proof of Prop. 4.1 (Appendix B.1) relies on setting the $\omega_\mathcal{N}$ of a GCCN to a simple, single-layer convolution.

2. **Permutation Equivariance.** Generalizing CC-NNs, GCCNs layers are equivariant with respect to the relabeling of cells in the combinatorial complex.

   **Proposition 4.2.** *A GCCN layer is cell permutation equivariant if the neighborhood message function is node permutation equivariant and the inter-neighborhood aggregator is cell permutation invariant.*

   Proof of Prop 4.2 (Appendix B.2) hinges on the node-wise permutation equivariance of $\omega_\mathcal{N}$ and the permutation invariance of the inter-neighborhood aggregation.

3. **Expressivity**. The expressiveness of TDL models is tied to their ability to distinguish non-isomorphic graphs. Variants of the Weisfeiler-Leman (WL) test, like the cellular WL for cell complexes (Bodnar et al., 2021a), set upper bounds on their corresponding TDL models' expressiveness, as the WL test does for GNNs (Xu et al., 2019b).

   **Proposition 4.3.** *GCCNs are strictly more expressive than CCNNs.*

   Proof of Prop. 4.3 (Appendix B.3) shows that GCCNs surpass CCNNs in expressivity by relating CCNNs to Weisfeiler-Leman (WL) and GCCNs to $k$-WL on augmented Hasse graphs.

---

Given Proposition 4.1, GCCNs allow us to define general TDL models using any neighborhood message function $\omega_\mathcal{N}$, such as any GNN. Not only does this framework avoid having to approximate CCNN computations, as is the case in previous works [4] (Jogl et al., 2022b;a; 2023), but it also enjoys the same permutation equivariance as regular CCNNs (Proposition 4.2). We show in Appendix C that the resulting time complexity of a GCCN is a compromise between a typical GNN and a CCNN. Differently from the work in (Hajij et al., 2023), the fact that GCCNs can have arbitrary neighborhood message functions implies that non message-passing TDL models can be readily defined. For example, one could choose $\omega_\mathcal{N}$ to be a spectral convolution neural network such as Defferrard et al. (2016). To the best of our knowledge, GCCNs are the only objects in the literature that encompass all the above properties.

## 5. TopoTune

Our proposed methodology, together with its resulting GC-CNs architectures, addresses the challenge of systematically generating principled, general TDL models. Here, we introduce TopoTune, a software module for defining and benchmarking GCCN architectures on the fly— a vehicle for accelerating TDL research. A quick start guide to the code and tutorial are provided at geometric-intelligence.github.io/topotune. This section details Topo-Tune's main features.

**Change of Paradigm.** TopoTune introduces a new perspective on TDL through the concept of "neighborhoods of interest," enabling unprecedented flexibility in architectural design. Previously fixed components of CCNNs, such as choice of topological domain, become hyperparameters of our framework.

**Accessible TDL.** Using TopoTune, a practitioner can instantiate customized GCCNs simply by modifying a few lines of a configuration file. In fact, it is sufficient to specify $(i)$ a collection of per-rank neighborhoods $\mathcal{N}_\mathcal{C}$, $(ii)$ a neighborhood message function $\omega_\mathcal{N}$, and optionally $(iii)$ some architectural parameters—e.g., the number $l$ of GCCN layers.[5] For the neighborhood message function $\omega_\mathcal{N}$, the same configuration file enables direct import of models from Py-Torch libraries such as PyTorch Geometric (Fey & Lenssen, 2019) and Deep Graph Library (Chen et al., 2020b). Topo-Tune's simplicity provides both newcomers and experts with an accessible tool for defining topological architectures.

---

[4] These models employ GNNs running on one augmented Hasse graph, i.e. a GCCN that, given a collection of neighborhoods $\mathcal{N}_\mathcal{C}$, uses a single neighborhood $\mathcal{N}_{tot}$ defined, for a cell $\sigma$, as $\mathcal{N}_{tot}(\sigma) = \bigcup_{\mathcal{N} \in \mathcal{N}_\mathcal{C}} \mathcal{N}(\sigma)$.

[5] We provide a detailed pseudo-code for TopoTune module in Appendix D.

**Accelerating TDL Research.** TopoTune is fully integrated into TopoBench (Telyatnikov et al., 2024), a comprehensive package offering a wide range of standardized methods and tools for TDL. Practitioners can access ready-to-use models, training pipelines, tasks, and evaluation metrics, including leading open-source models from TopoX (Hajij et al., 2024b). In addition, TopoBench features the largest collection of *topological liftings* currently available—transformations that map graph datasets into higher-order topological domains. Together, TopoBench and TopoTune organize the vast design space of TDL into an accessible framework, providing unparalleled versatility and standardization for practitionners.

# 6. Experiments

We present experiments showcasing a broad class of GCCN's constructed with TopoTune. These models consistently match, outperform, or finetune existing CCNNs, often with smaller model sizes. TopoTune's integration into the TopoBench experiment infrastructure ensures a fair comparison with CCNNs from the literature, as data processing, domain lifting, and training are homogeonized.

## 6.1. Experimental Setup

We generate our class of GCCNs by considering ten possible choices of neighborhood structure $\mathcal{N}_\mathcal{C}$ (including both regular and per-rank, see Appendix E.1) and five possible choices of $\omega_\mathcal{N}$: GCN (Kipf & Welling, 2017), GAT (Velickovic et al., 2017), GIN (Xu et al., 2019a), GraphSAGE (Hamilton et al., 2017), and Transformer (Vaswani et al., 2017). We import these models directly from PyTorch Geometric (Fey & Lenssen, 2019) and PyTorch (Paszke et al., 2019). TopoTune enables running GCCNs on both an ensemble of strictly augmented Hasse graphs (eq. 6) and an augmented Hasse graph (eq. 5). While CCNN results reflect extensive hyperparameter tuning by Telyatnikov et al. (2024) (see that work's Appendix C.2 for details), we largely fix GCCN training hyperparameters using the TopoBench default configuration.

**Datasets.** We include a wide range of benchmark tasks (see Appendix E.2). MUTAG, PROTEINS, NCI01, and NCI09 (Morris et al., 2020) are graph-level classification tasks about molecules or proteins. ZINC (Irwin et al., 2012) (subset) is a graph-level regression task about solubility. At the node level, the Cora, CiteSeer, and PubMed tasks (Yang et al., 2016) involve classifying publications within citation networks. We consider two topological domains: simplicial and cellular complexes. We use TopoBench's lifting processes to infer higher-order relationships in these data. We use node features to construct edge and face features.

## 6.2. Results and Discussion

**GCCNs outperform CCNNs.** Table 1 compares top-performing CCNNs with our class of GCCNs. GCCNs outperform CCNNs across all datasets in the simplicial and cellular domains and match hypergraph CCNNs—something CCNNs fail to achieve in node-level tasks. Across 16 domain/dataset combinations, GCCNs exceed the best CCNN by $> 1\sigma$ in 11 cases. In 2 combinations, GCCNs are outperformed by the best GNN included in Telyatnikov et al. (2024). Representing complexes as ensembles of augmented Hasse graphs, rather than a single graph, improves results.

**Generalizing existing CCNNs to GCCNs improves performance.** TopoTune makes it easy to iterate upon and improve preexisting CCNNs by replicating their architecture in a GCCN setting. For example, TopoTune can generate a counterpart GCCN by replicating a CCNN's neighborhood structure, aggregation, and training scheme. We show in Table 2 that counterpart GCCNs can achieve comparable or better results than SCCN (Yang et al., 2022) and CWN (Bodnar et al., 2021a) just by sweeping over additional choices of $\omega_\mathcal{N}$. In the single augmented Hasse graph regime, GCCN models are consistently more lightweight, up to half their size (see Table 5).

**GCCNs perform competitively to CCNNs with fewer parameters.** GCCNs are often more parameter efficient than existing CCNNs in simplicial and cellular domains, and in some instances (MUTAG, NCI1, NCI09), even in the hypergraph domain. We refer to Table 4. Even as GCCNs become more parameter-intensive for large graphs with high-dimensional embeddings—as seen in node-level tasks—they remain competitive. (We refer to Appendix H.1 for additional results on larger node-level datasets.) For instance, on the Citeseer dataset, a GCCN ($\omega_\mathcal{N}$ = GraphSAGE) outperforms the best existing CCNN while being 28% smaller. Training times in Appendix G show GCCNs train comparably on smaller datasets but slow down on larger ones, likely due to TopoTune's on-the-fly graph expansion. Preprocessing this expansion in future work could mitigate the lag.

**TopoTune finds parameter-efficient GCCNs.** By easily exploring a wide landscape of possible GCCNs for a given task, TopoTune helps identify models that maximize performance while minimizing model size. Fig. 5 illustrates this trade-off by comparing the performance and size of selected GCCNs (see Appendix I for more). On the PROTEINS dataset, two GCCNs using per-rank neighborhood structures (orange and black) achieve performance within 2% of the best result while requiring as little as 48% of the parameters. This reduction is due to fewer neighborhoods $\mathcal{N}$, resulting in fewer $\omega_\mathcal{N}$ blocks per GCCN layer. Similarly,

*Table 1.* Cross-domain, cross-task, cross-expansion, and cross-$\omega_{\mathcal{N}}$ comparison of GCCN architectures with top-performing CCNNs benchmarked on TopoBench (Telyatnikov et al., 2024). Best result is in **bold** and results within 1 standard deviation are highlighted blue . Experiments are run with 5 seeds. We report accuracy for classification tasks and MAE for regression.

| Model | Graph-Level Tasks | | | | | Node-Level Tasks | | |
|---|---|---|---|---|---|---|---|---|
| | MUTAG (↑) | PROTEINS (↑) | NCI1 (↑) | NCI109 (↑) | ZINC (↓) | Cora (↑) | Citeseer (↑) | PubMed (↑) |
| Graph | | | | | | | | |
| GNN (Best Model on TopoBench) | 79.57 ± 6.13 | 76.34 ± 1.66 | 75.00 ± 0.99 | 74.42 ± 0.70 | 0.57 ± 0.04 | 87.21 ± 1.89 | 75.53 ± 1.27 | 89.44 ± 0.24 |
| Cellular | | | | | | | | |
| CCNN (Best Model on TopoBench) | 80.43 ± 1.78 | 76.13 ± 2.70 | 76.67 ± 1.48 | 75.35 ± 1.50 | 0.34 ± 0.01 | 87.44 ± 1.28 | 75.63 ± 1.58 | 88.64 ± 0.36 |
| GCCN $\omega_{\mathcal{N}}$ = GAT | 83.40 ± 4.85 | 74.05 ± 2.16 | 76.11 ± 1.69 | 75.62 ± 0.76 | 0.38 ± 0.03 | 88.39 ± 0.65 | 74.62 ± 1.95 | 87.68 ± 0.33 |
| GCCN $\omega_{\mathcal{N}}$ = GCN | 85.11 ± 6.73 | 74.41 ± 1.77 | 76.42 ± 1.67 | 75.62 ± 0.94 | 0.36 ± 0.01 | 88.51 ± 0.70 | 75.41 ± 2.00 | 88.18 ± 0.26 |
| GCCN $\omega_{\mathcal{N}}$ = GIN | **86.38 ± 6.49** | 72.54 ± 3.07 | 77.65 ± 1.11 | **77.19 ± 0.21** | **0.19 ± 0.00** | 87.42 ± 1.85 | 75.13 ± 1.17 | 88.47 ± 0.27 |
| GCCN $\omega_{\mathcal{N}}$ = GraphSAGE | 85.53 ± 6.80 | 73.62 ± 2.72 | **78.23 ± 1.47** | 77.10 ± 0.83 | 0.24 ± 0.00 | 88.57 ± 0.58 | 75.89 ± 1.84 | 89.40 ± 0.57 |
| GCCN $\omega_{\mathcal{N}}$ = Transformer | 83.83 ± 6.49 | 70.97 ± 4.06 | 73.00 ± 1.37 | 73.20 ± 1.05 | 0.45 ± 0.02 | 84.61 ± 1.32 | 75.05 ± 1.67 | 88.37 ± 0.22 |
| GCCN $\omega_{\mathcal{N}}$ = Best GNN, 1 Aug. Hasse graph | 85.96 ± 7.15 | 73.73 ± 2.95 | 76.75 ± 1.63 | 76.94 ± 0.82 | 0.31 ± 0.01 | 87.24 ± 0.58 | 74.26 ± 1.47 | 88.65 ± 0.55 |
| Simplicial | | | | | | | | |
| CCNN (Best Model on TopoBench) | 76.17 ± 6.63 | 75.27 ± 2.14 | 76.60 ± 1.75 | 77.12 ± 1.07 | 0.36 ± 0.02 | 82.27 ± 1.34 | 71.24 ± 1.68 | 88.72 ± 0.50 |
| GCCN $\omega_{\mathcal{N}}$ = GAT | 79.15 ± 4.09 | 74.62 ± 1.95 | 74.86 ± 1.42 | 74.81 ± 1.14 | 0.57 ± 0.03 | 88.33 ± 0.67 | 74.65 ± 1.93 | 87.72 ± 0.36 |
| GCCN $\omega_{\mathcal{N}}$ = GCN | 74.04 ± 8.30 | 74.91 ± 2.51 | 74.20 ± 2.17 | 74.13 ± 0.53 | 0.53 ± 0.05 | 88.51 ± 0.70 | 75.41 ± 2.00 | 88.19 ± 0.24 |
| GCCN $\omega_{\mathcal{N}}$ = GIN | 85.96 ± 4.66 | 72.83 ± 2.72 | 76.67 ± 1.62 | 75.76 ± 1.28 | 0.35 ± 0.01 | 87.27 ± 1.63 | 75.05 ± 1.27 | 88.54 ± 0.21 |
| GCCN $\omega_{\mathcal{N}}$ = GraphSAGE | 75.74 ± 2.43 | 74.70 ± 3.10 | 76.85 ± 1.50 | 75.64 ± 1.94 | 0.50 ± 0.02 | 88.57 ± 0.59 | **75.92 ± 1.85** | 89.34 ± 0.39 |
| GCCN $\omega_{\mathcal{N}}$ = Transformer | 74.04 ± 4.09 | 70.97 ± 4.06 | 70.39 ± 0.96 | 69.99 ± 1.13 | 0.64 ± 0.01 | 84.4 ± 1.16 | 74.6 ± 1.88 | 88.55 ± 0.39 |
| GCCN $\omega_{\mathcal{N}}$ = Best GNN, 1 Aug. Hasse graph | 74.04 ± 5.51 | 74.48 ± 1.89 | 75.02 ± 2.24 | 73.91 ± 3.9 | 0.56 ± 0.02 | 87.56 ± 0.66 | 74.5 ± 1.61 | 88.61 ± 0.27 |
| Hypergraph | | | | | | | | |
| CCNN (Best Model on TopoBench) | 80.43 ± 4.09 | **76.63 ± 1.74** | 75.18 ± 1.24 | 74.93 ± 2.50 | 0.51 ± 0.01 | **88.92 ± 0.44** | 74.93 ± 1.39 | **89.62 ± 0.25** |

*Table 2.* We compare existing CCNNs with $\omega_{\mathcal{N}}$-modified GCCN counterparts. We show the result for best choice of $\omega_{\mathcal{N}}$. Experiments are run with 5 seeds.

| Model | Graph-Level Tasks | | | | Node-Level Tasks | | |
|---|---|---|---|---|---|---|---|
| | MUTAG | PROTEINS | NCI1 | NCI109 | Cora | Citeseer | PubMed |
| SCCN (Yang et al., 2022) | | | | | | | |
| Benchmark results (Telyatnikov et al., 2024) | 70.64 ± 5.90 | 74.19 ± 2.86 | **76.60 ± 1.75** | **77.12 ± 1.07** | 82.19 ± 1.07 | 69.60 ± 1.83 | **88.18 ± 0.32** |
| GCCN, on ensemble of strictly aug. Hasse graphs | **82.13 ± 4.66** | 75.56 ± 2.48 | 75.6 ± 1.28 | 74.19 ± 1.44 | **88.06 ± 0.93** | 74.67 ± 1.24 | 87.70 ± 0.19 |
| GCCN, on 1 aug. Hasse graph | 69.79 ± 4.85 | 74.48 ± 2.67 | 74.63 ± 1.76 | 70.71 ± 5.50 | 87.62 ± 1.62 | **74.86 ± 1.7** | 87.80 ± 0.28 |
| CWN (Bodnar et al., 2021a) | | | | | | | |
| Benchmark results (Telyatnikov et al., 2024) | 80.43 ± 1.78 | **76.13 ± 2.70** | 73.93 ± 1.87 | 73.80 ± 2.06 | 86.32 ± 1.38 | **75.20 ± 1.82** | **88.64 ± 0.36** |
| GCCN, on ensemble of strictly aug. Hasse graphs | **84.26 ± 8.19** | 75.91 ± 2.75 | 73.87 ± 1.10 | 73.75 ± 0.49 | 85.64 ± 1.38 | 74.89 ± 1.45 | 88.40 ± 0.46 |
| GCCN, on 1 aug. Hasse graph | 81.70 ± 5.34 | 75.05 ± 2.39 | **75.14 ± 0.76** | **75.39 ± 1.01** | **86.44 ± 1.33** | 74.45 ± 1.59 | 88.56 ± 0.55 |

on ZINC, lightweight neighborhood structures (orange and dark green) are competitive with reduced parameter costs. Node-level tasks, see less benefit, likely due to the larger graph sizes and higher-dimensional input features.

**Impactfulness of GNN choice is dataset specific.** Fig. 5 also provides insights into the impact of neighborhood message functions. On ZINC, GIN clearly outperforms all other models, which do not even appear in the plot's range. In the less clear-cut cases of PROTEINS and Citeseer, we observe a trade-off between neighborhood structure and message function complexity. We find that larger base models (GIN, GraphSAGE) on lightweight neighborhood structures perform comparably to simpler base models (GAT, GCN) on larger neighborhood structures. This tradeoff warrants further research on the dataset-specific importance of neighborhood choice, or lack thereof. We refer to Appendix H.2 for additional experiments with more advanced GNNs,

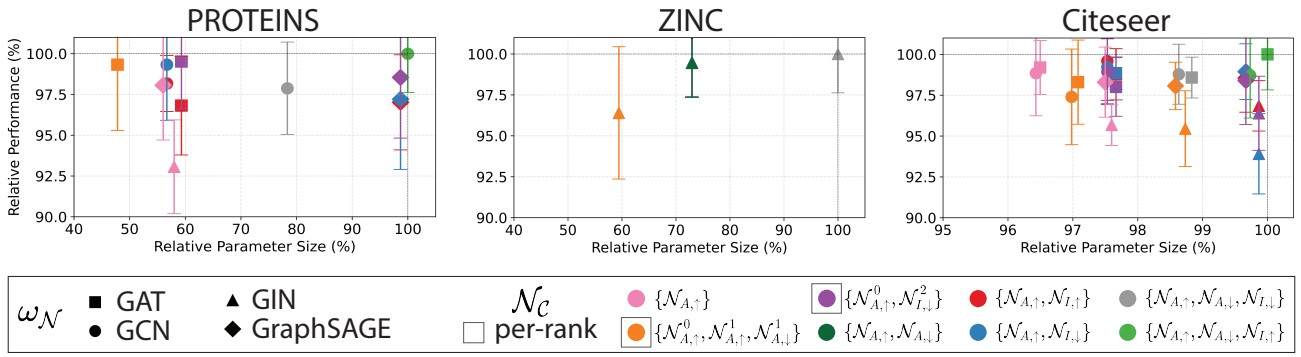

*Figure 5.* **GCCN performance versus size.** We compare various GCCNs across three datasets on the cellular domain, two graph-level (left, middle) and one node-level (right). Each GCCN (point) has a different neighborhood structure $\mathcal{N}_\mathcal{C}$, some of which can only be represented as per-rank structures ($\square$ in legend), and message function $\omega_\mathcal{N}$. The amount of layers is kept constant according to the best performing model. The axes are scaled relative to this model.

GATv2 (Brody et al., 2021) and PNA (Corso et al., 2020), as choices of $\omega_\mathcal{N}$.

## 7. Conclusion

This work introduces a simple yet powerful graph-based methodology for constructing Generalized Combinatorial Complex Neural Networks (GCCNs), TDL architectures that generalize and subsume standard CCNNs. Additionally, we introduce TopoTune, the first lightweight software module for systematically and easily implementing new TDL architectures across topological domains. In doing so, we have addressed, either in part or in full, 7 of the 11 open problems of the field (Papamarkou et al., 2024). Future work includes customizing GCCNs for application-specific and potentially sparse or multimodal datasets, and leveraging software from state-of-the-art GNNs. TopoTune will also help bridge the gap with other fields such as attentional learning and $k$-hop higher-order GNNs (Morris et al., 2019; Maron et al., 2019).

## Acknowledgements

M.P. acknowledges the support of National Science Foundation (NSF) CAREER 2240158 and NSF Grant 2134241, as well as from the National Science and Engineering Research Council of Canada. G.B. acknowledges support from NSF Grant 2134241. C.B. acknowledges support from the National Institutes of Health Grant 1R01ES037156-01. N.M. acknowledges support from NSF Grant 2313150.

## Impact Statement

This paper presents work whose goal is to advance the field of Machine Learning. There are many potential societal consequences of our work, none which we feel must be specifically highlighted here.

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

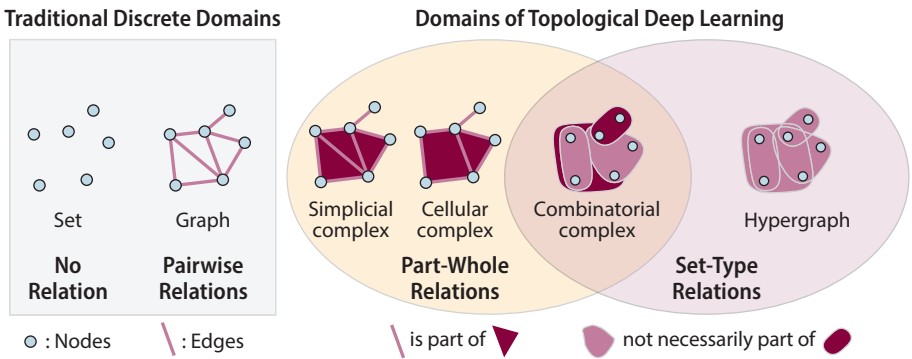

*Figure 6.* **Topological Deep Learning Domains.** Nodes in blue, (hyper)edges in pink, and faces in dark red. Figure adopted from Papillon et al. (2023).

## A. Domains of Topological Deep Learning

We summarize the different discrete domains leveraged within TDL and, in doing so, contextualize how combinatorial complexes generalize all of them. To that end, we will closely follow the description of (Papillon et al., 2023), using as well its very clarifying Figure 6. We recommend this survey for a high-level overview of TDL literature, and the more extensive work of (Hajij et al., 2023) for a detailed formulation of the field. We also refer to Appendix C of Battiloro et al. (2025) for a concise mathematical description of each domain. From left to right in Figure 6, the different domains in TDL are:

**Traditional Discrete Domains**

**Set / Pointcloud.**    A collection of points called *nodes* without any additional structure.

**Graph.**    A set of points (nodes) connected with edges that denote pairwise relationships.

**Set + Part-Whole Relations**

**Simplicial Complex.**    A generalization of a graph that incorporates hierarchical part-whole relations through the multi-scale construction of cells. Nodes are rank 0-cells that can be combined to form edges (rank 1 cells). Edges are, in turn, combined to form faces (rank 2 cells), which are combined to form volumes (rank 3 cells), and so on. In particular, each cell $\sigma$ in a simplicial complex must contain all lower dimensional cells $\tau$ such that $\tau \subseteq \sigma$. Therefore, faces must be triangles, volumes must be tetrahedrons, and so forth.

**Cellular Complex.**    A generalization of an simplicial complex in which cells are not limited to simplexes, but may instead take any shape: faces can involve more than three nodes, volumes more than four faces, and so on. This flexibility endows CCs with greater expressivity than simplicial complexes (Bodnar et al., 2021a), but still edges only connect pairs of nodes.

**Set + Set-Type Relations**

**Hypergraph:**    A generalization of a graph, in which higher-order edges called hyperedges can connect arbitrary sets of two or more nodes. Connections in HGs represent set-type relationships, in which participation in an interaction is not implied by any other relation in the system. This makes HGs an ideal choice for data with abstract and arbitrarily large interactions of equal importance, such as semantic text and citation networks.

**Set + Part-Whole and Set-Type Relations**

**Combinatorial Complex:**    A structure that combines features of hypergraphs and cellular complexes. Like a hypergraph, edges may connect any number of nodes. Like a cellular complex, cells can be combined to form higher-ranked structures. Hence, combinatorial complexes generalize all other topological domains.

# B. Proofs

## B.1. Proof of Generality

The proof is straightforward. It is sufficient to set $\omega_{\mathcal{N}}(\mathbf{H}_{\mathcal{N}}^l, \mathcal{G}_{\mathcal{N}})$ to $\{\bigoplus_{y \in \mathcal{N}(\sigma)} \psi_{\mathcal{N},\mathrm{rk}(\sigma)} (\mathbf{h}_\sigma^l, \mathbf{h}_\tau^l)\}_{\sigma \in \mathcal{C}}$ in (8) as all $y \in \mathcal{N}(\sigma)$ are part of the node set $\mathcal{C}_{\mathcal{N}}$ of the strictly augmented Hasse graph of $\mathcal{N}$ by definition.

## B.2. Proof of Equivariance

As for GNNs, an amenable property for GCCNNs is the awareness w.r.t. relabeling of the cells. In other words, given that the order in which the cells are presented to the networks is arbitrary -because CCs, like (undirected) graphs, are purely combinatorial objects-, one would expect that if the order changes, the output changes accordingly. To formalize this concept, we need the following notions.

**Matrix Representation of a Neighborhood.** Assume again to have a combinatorial complex $\mathcal{C}$ containing $C := |\mathcal{C}|$ cells and a neighborhood function $\mathcal{N}$ on it. Assume again to give an arbitrary labeling to the cells in the complex, and denote the $i$-th cell with $\sigma_i$. The matrix representation of the neighborhood function is a matrix $\mathbf{N}_{\mathcal{N}} \in \mathbb{R}^{C \times C}$ such that $\mathbf{N}_{i,j} = 1$ if the $\sigma_j \in \mathcal{N}(\sigma_i)$ or zero otherwise. We notice that the submatrix $\widetilde{\mathbf{N}}_{\mathcal{N}} \in \mathbb{R}^{|\mathcal{C}_{\mathcal{N}}| \times |\mathcal{C}_{\mathcal{N}}|}$ obtained by removing all the zero rows and columns is the adjacency matrix of the strictly augmented Hasse graph $\mathcal{G}_{\mathcal{C}_{\mathcal{N}}}$ induced by $\mathcal{N}$.

**Permutation Equivariance.** Let $\mathcal{C}$ be combinatorial complex, $\mathcal{N}_{\mathcal{C}}$ a collection of neighborhoods on it, and $\mathbf{N} = \{\mathbf{N}_{\mathcal{N}}\}_{\mathcal{N} \in \mathcal{N}_{\mathcal{C}}}$ the set collecting the corresponding neighborhood matrices. Let $\mathbf{P} \in \mathbb{R}^{C \times C}$ be a permutation matrix. Finally, denote by $\mathbf{PH}$ the permuted embeddings and by $\{\mathbf{PN}_{\mathcal{N}}\mathbf{P}^T\}_{\mathcal{N} \in \mathcal{N}_{\mathcal{C}}}$, the permuted neighborhood matrices. We say that a function $f : (\mathbf{H}^l, \mathbf{B}) \mapsto \mathbf{H}^{l+1}$ is cell permutation equivariant if $f\left(\mathbf{PH}^l, \{\mathbf{PN}_{\mathcal{N}}\mathbf{P}^T\}_{\mathcal{N} \in \mathcal{N}_{\mathcal{C}}}\right) = \mathbf{P}f\left(\mathbf{H}^l, \{\mathbf{N}_{\mathcal{N}}\}_{\mathcal{N} \in \mathcal{N}_{\mathcal{C}}}\right)$ for any permutation matrix $\mathbf{P}$. Intuitively, the permutation matrix changes the arbitrary labeling of the cells, and a permutation equivariant function is a function that reflects the change in its output.

*Proof of Proposition 4.2.* We follow the approach from (Bodnar et al., 2021a). Given any permutation matrix $\mathbf{P}$, for a cell $\sigma_i$, let us denote its permutation as $\sigma_{\mathbf{P}(i)}$ with an abuse of notation. Let $\mathbf{h}_{\sigma_i}^{l+1}$ be the output embedding of cell $\sigma_i$ for the $l$-th layer of a GCCN taking $(\mathbf{H}^l, \{\mathbf{N}_{\mathcal{N}}\}_{\mathcal{N} \in \mathcal{N}_{\mathcal{C}}})$ as input, and $\mathbf{h}_{\sigma_{\mathbf{P}(i)}}^{l+1}$ be the output embedding of cell $\sigma_{\mathbf{P}(i)}$ for the same GCCN layer taking $\left(\mathbf{PH}^l, \{\mathbf{PN}_{\mathcal{N}}\mathbf{P}^T\}_{\mathcal{N} \in \mathcal{N}_{\mathcal{C}}}\right)$ as input. To prove the permutation equivariance, it is sufficient to show that $\mathbf{h}_{\sigma_i}^{l+1} = \mathbf{h}_{\sigma_{\mathbf{P}(i)}}^{l+1}$ as the update function $\phi$ is row-wise, i.e., it independently acts on each cell. To do so, we show that the (multi-)set of embeddings being passed to the neighborhood message function, aggregation, and update functions are the same for the two cells $\sigma_i$ and $\sigma_{\mathbf{P}(i)}$. The neighborhood message functions act on the strictly augmented Hasse graph of $\mathcal{G}_{\mathcal{C}_{\mathcal{N}}}$ of $\mathcal{N}$, thus we work with the submatrix $\widetilde{\mathbf{N}}_{\mathcal{N}}$. The neighborhood message function is assumed to be *node* permutation equivariant, i.e., denoting again the embeddings of the cells in $\mathcal{G}_{\mathcal{C}_{\mathcal{N}}}$ with $\mathbf{H}_{\mathcal{C}_{\mathcal{N}}}^l \in \mathbb{R}^{|\mathcal{C}_{\mathcal{N}}| \times F^l}$ and identifying $\mathcal{G}_{\mathcal{C}_{\mathcal{N}}}$ with $\widetilde{\mathbf{N}}_{\mathcal{N}}$, it holds that $\omega_{\mathcal{N}}(\mathbf{P}_{\mathcal{C}_{\mathcal{N}}}\mathbf{H}_{\mathcal{C}_{\mathcal{N}}}^l, \mathbf{P}_{\mathcal{C}_{\mathcal{N}}}\widetilde{\mathbf{N}}_{\mathcal{N}}\mathbf{P}_{\mathcal{C}_{\mathcal{N}}}^T) = \mathbf{P}_{\mathcal{C}_{\mathcal{N}}}\omega_{\mathcal{N}}(\mathbf{H}_{\mathcal{C}_{\mathcal{N}}}^l, \widetilde{\mathbf{N}}_{\mathcal{N}})$, where $\mathbf{P}_{\mathcal{C}_{\mathcal{N}}}$ is the submatrix of $\mathbf{P}$ given by the rows and the columns corresponding to the cells in $\mathcal{G}_{\mathcal{C}_{\mathcal{N}}}$. This assumption, together with the assumption that the inter-neighborhood aggregation is assumed to be *cell* permutation invariant, i.e. $\bigotimes_{\mathcal{N} \in \mathcal{N}_{\mathcal{C}}} \mathbf{P}_{\mathcal{C}_{\mathcal{N}}}\omega_{\mathcal{N}}(\mathbf{H}_{\mathcal{C}_{\mathcal{N}}}^l, \widetilde{\mathbf{N}}_{\mathcal{N}}) = \bigotimes_{\mathcal{N} \in \mathcal{N}_{\mathcal{C}}} \omega_{\mathcal{N}}(\mathbf{H}_{\mathcal{C}_{\mathcal{N}}}^l, \widetilde{\mathbf{N}}_{\mathcal{N}})$, trivially makes the overall composition of the neighborhood message function with the inter-neighborhood aggregation *cell* permutation invariant. This fact, together with the fact that the (labels of) the neighbors of the cell $\sigma_i$ in $\mathcal{N}$ are given by the nonzero elements of the $i$-th row of $\mathbf{N}_{\mathcal{N}}$, or the corresponding row of $\widetilde{\mathbf{N}}_{\mathcal{N}}$, and that the columns and rows of $\widetilde{\mathbf{N}}_{\mathcal{N}}$ are permuted in the same way the rows of the feature matrix $\mathbf{H}_{\mathcal{C}_{\mathcal{N}}}^l$ are permuted, implies

$$[\widetilde{\mathbf{N}}_{\mathcal{N}}]_{i,j} = [\mathbf{P}_{\mathcal{C}_{\mathcal{N}}}\widetilde{\mathbf{N}}_{\mathcal{N}}\mathbf{P}_{\mathcal{C}_{\mathcal{N}}}^T]_{\mathbf{P}_{\mathcal{C}_{\mathcal{N}}}(i), \mathbf{P}_{\mathcal{C}_{\mathcal{N}}}(j)}, \tag{9}$$

thus that $\sigma_i$ and $\sigma_{\mathbf{P}(i)}$ receive the same neighborhood message from the neighboring cells in $\mathcal{N}$, for all $\mathcal{N} \in \mathcal{N}_{\mathcal{C}}$.

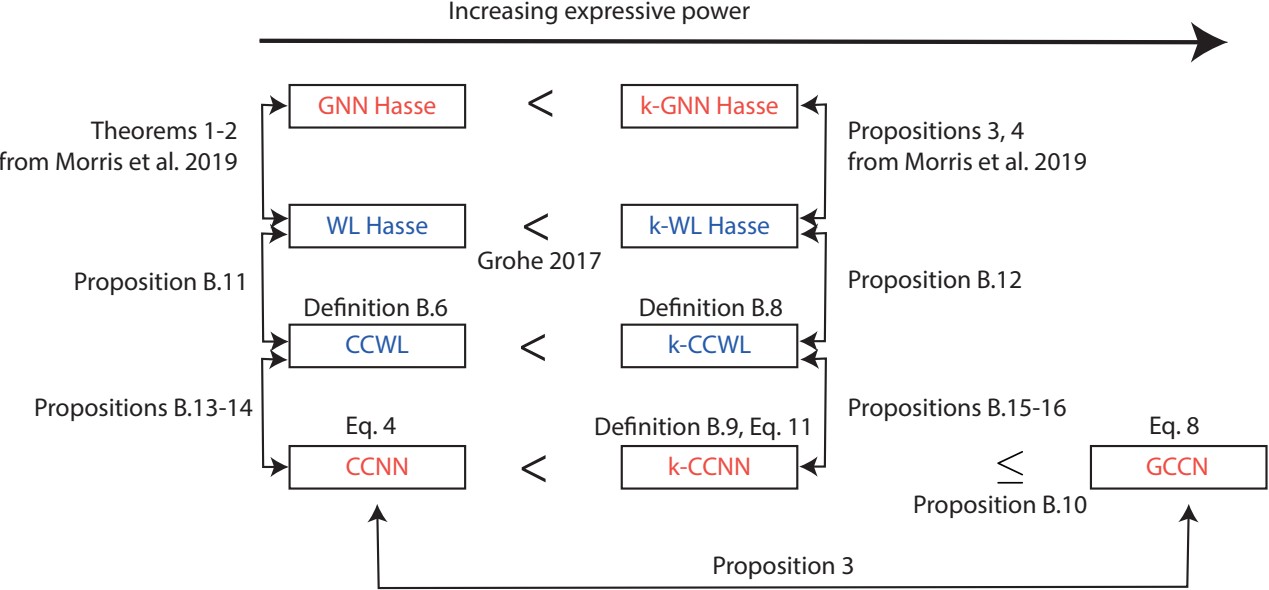

*Figure 7.* Graphical summary of the definitions and propositions related to the expressivity of CCNNs and GCCNs and of the different WL tests. Neural networks expressivity is in red, and WL test expressivity is in blue.

## B.3. Proof of Expressivity

We provide the theory required to prove Proposition 4.3, i.e., to prove that GCCNs are strictly more expressive than CCNNs. The definitions and propositions from this subsection are summarized in Figure 7. This figure serves as a graphical reading guide for the subsection.

### B.3.1. HOMOMORPHISM AND ISOMORPHISM INDUCED BY NEIGHBORHOODS

We first recall the notion of homomorphism of a combinatorial complex (CC) from (Hajij et al., 2023) and generalize it to the notions of homomorphism and isomorphism of CCs induced by a neighborhood $\mathcal{N}$.

**Definition B.1** (CC-Homomorphism (Hajij et al., 2023)). A homomorphism from a CC $(\mathcal{V}_1, \mathcal{C}_1, \mathrm{rk}_1)$ to a CC $(\mathcal{V}_2, \mathcal{C}_2, \mathrm{rk}_2)$, also called a CC-homomorphism, is a function $f : \mathcal{C}_1 \to \mathcal{C}_2$ that satisfies the following conditions:

1. If $\sigma, \tau \in \mathcal{C}_1$ satisfy $\sigma \subseteq \tau$, then $f(\sigma) \subseteq f(\tau)$.

2. If $\sigma \in \mathcal{C}_1$, then $\mathrm{rk}_1(\sigma) \geq \mathrm{rk}_2(f(\sigma))$.

Definition B.1 proposes a CC-homomorphism that respects the incidence structures of the CCs, denoted by the symbol $\subseteq$ in the definition above. We generalize Definition B.1 by allowing CC-homomorphisms to take into account a labeling of the cells and to be defined in terms of general neighborhood structures beyond incidence. We first define a labeled combinatorial complex.

**Definition B.2** (Labeled Combinatorial Complex). A labeled combinatorial complex $(\mathcal{C}, \ell)$ is a CC $\mathcal{C}$ equipped with a cell coloring $\ell : \mathcal{C} \mapsto \Sigma$ with arbitrary codomain $\Sigma$. We say that $\ell(\sigma)$ is a label or color of cell $\sigma \in \mathcal{C}$.

Next, we provide our definitions of homomorphisms.

**Definition B.3** (CC-Homomorphism induced by $(\mathcal{N}_1, \mathcal{N}_2)$). A homomorphism from a CC $(\mathcal{V}_1, \mathcal{C}_1, \mathrm{rk}_1)$ with neighborhood $\mathcal{N}_1$ to a CC $(\mathcal{V}_2, \mathcal{C}_2, \mathrm{rk}_2)$ with neighborhood $\mathcal{N}_2$, also called a CC-homomorphism induced by $(\mathcal{N}_1, \mathcal{N}_2)$, is a function $f : \mathcal{C}_1 \to \mathcal{C}_2$ that satisfies: If $\sigma, \tau \in \mathcal{C}_1$ are such that $\tau \in \mathcal{N}_1(\sigma)$, then $f(\tau) \in \mathcal{N}_2(f(\sigma))$. A labeled CC-homomorphism induced by $(\mathcal{N}_1, \mathcal{N}_2)$ is a CC-homomorphism induced by $(\mathcal{N}_1, \mathcal{N}_2)$ that additionally respects labeling of the cells, that is: if $\sigma, \tau \in \mathcal{C}_1$ have the same label, then $f(\sigma), f(\tau) \in \mathcal{C}_2$ also have the same label.

We prove that a CC-homomorphism induced by $(\mathcal{N}_1, \mathcal{N}_2)$ is equivalent to a homomorphism of the respective strictly augmented Hasse graphs $\mathcal{G}_{\mathcal{N}_1}$, $\mathcal{G}_{\mathcal{N}_2}$.

**Proposition B.4.** *For every CC-homomorphism $f$ from $\mathcal{C}_1$ to $\mathcal{C}_2$ induced by $(\mathcal{N}_1, \mathcal{N}_2)$, there exists a unique graph homomorphism between their respective strictly augmented Hasse graphs $\mathcal{G}_{\mathcal{N}_1}$ and $\mathcal{G}_{\mathcal{N}_2}$.*

*Proof.* Consider $f$ a CC-homomorphism from $\mathcal{C}_1$ to $\mathcal{C}_2$ induced by $(\mathcal{N}_1, \mathcal{N}_2)$ as in Definition B.3. Define the function $\tilde{f}$ from nodes of $\mathcal{G}_{\mathcal{N}_1}$ to the nodes of $\mathcal{G}_{\mathcal{N}_2}$ corresponding to $f$, i.e., $\tilde{f} : \mathcal{C}_{\mathcal{N}_1} \mapsto \mathcal{C}_{\mathcal{N}_2}$ defined as $\tilde{f}(\tilde{\sigma}) = \widetilde{f(\sigma)}$ where $\tilde{\sigma}$ is the node in $\mathcal{G}_{\mathcal{N}_1}$ corresponding to the cell $\sigma$ in $\mathcal{C}_1$, and $\widetilde{f(\sigma)}$ is the node in $\mathcal{G}_{\mathcal{N}_2}$ corresponding to the cell $f(\sigma)$ in $\mathcal{C}_2$. We show that $\tilde{f}$ is a graph homomorphism from $\mathcal{G}_{\mathcal{N}_1}$ to $\mathcal{G}_{\mathcal{N}_2}$, i.e., a function from the nodes of $\mathcal{G}_{\mathcal{N}_1}$ to the nodes of $\mathcal{G}_{\mathcal{N}_2}$ that preserves edges.

By definition of the CC-homomorphism induced by $(\mathcal{N}_1, \mathcal{N}_2)$, we have: if $\tau \in \mathcal{N}_1(\sigma)$ then $f(\tau) \in \mathcal{N}_2(f(\sigma))$. Recognizing that $\mathcal{N}_1$ defines edges of $\mathcal{G}_{\mathcal{N}_1}$, and $\mathcal{N}_2$ defines edgess of $\mathcal{G}_{\mathcal{N}_2}$, we have: if $(\tilde{\sigma}, \tilde{\tau})$ is an edge in $\mathcal{G}_{\mathcal{N}_1}$, then $(\tilde{f}(\tilde{\sigma}), \tilde{f}(\tilde{\tau}))$ is an edge in $\mathcal{G}_{\mathcal{N}_2}$. Thus, a CC-homomorphism induced by $(\mathcal{N}_1, \mathcal{N}_2)$ gives a homomorphism of the strictly augmented Hasse graphs.

Conversely, if $\tilde{f}$ is a graph homomorphism from $\mathcal{G}_{\mathcal{N}_1}$ to $\mathcal{G}_{\mathcal{N}_2}$, then we similarly construct a CC-homomorphism $f$ between $\mathcal{C}_1$ and $\mathcal{C}_2$. This concludes the proof. $\qquad\square$

Lastly, we can define a notion of CC-isomorphism induced by neighborhood structures.

**Definition B.5** (CC-Isomorphism induced by $(\mathcal{N}_1, \mathcal{N}_2)$). A isomorphism from a CC $(\mathcal{V}_1, \mathcal{C}_1, \mathrm{rk}_1)$ with neighborhood $\mathcal{N}_1$ to a CC $(\mathcal{V}_2, \mathcal{C}_2, \mathrm{rk}_2)$ with neighborhood $\mathcal{N}_2$, also called a CC-isomorphism induced by $(\mathcal{N}_1, \mathcal{N}_2)$, is an invertible CC-homomorphism induced by $(\mathcal{N}_1, \mathcal{N}_2)$ whose inverse is a CC-isomorphism induced by $(\mathcal{N}_2, \mathcal{N}_1)$. A labeled CC-isomorphism induced by $(\mathcal{N}_1, \mathcal{N}_2)$ is a CC-isomorphism that additionally respects labels.

### B.3.2. WEISFEILER-LEMAN (WL) TESTS ON COMBINATORIAL COMPLEXES

We propose two WL tests, called CCWL and (set-based) $k$-CCWL that generalize the WL and the (set-based) $k$-WL tests to labeled combinatorial complexes. We start with the generalization of the WL test to labeled combinatorial complexes.

**Definition B.6** (The CC Weisfeiler-Leman (CCWL) test on labeled combinatorial complexes). Let $(\mathcal{C}, \ell)$ be a labeled combinatorial complex. Let $\mathcal{N}$ be a neighborhood on $\mathcal{C}$. The scheme proceeds as follows:

- Initialization: Cells $\sigma$ are initialized with the labels given by $\ell$, i.e.: for all $\sigma \in \mathcal{C}$, we set: $c_{\sigma,\ell}^0 = \ell(\sigma)$.

- Refinement: Given colors of cells at iteration $t$, the refinement step computes the color of cell $\sigma$ at the next iteration $c_{\sigma,\ell}^{t+1}$ using a perfect HASH function as follows:

$$c_{\mathcal{N}}^t(\sigma) = \left\{\!\left\{ c_{\sigma',\ell}^t \mid \forall \sigma' \in \mathcal{N}(\sigma) \right\}\!\right\},$$
$$c_{\sigma,\ell}^{t+1} = \mathrm{HASH}\left( c_{\sigma,\ell}^t, c_{\mathcal{N}}^t(\sigma) \right).$$

- Termination: The algorithm stops when an iteration leaves the coloring unchanged.

Next, we generalize the set-based $k$-WL test to labeled combinatorial complexes, called the $k$-CCWL test. The set-based $k$-WL test is employed in (Morris et al., 2019) where colors are defined on $k$-sets of nodes, as opposed to $k$-tuples of nodes in the standard $k$-WL test. Specifically, we denote $[\mathcal{C}]^k$ the set of $k$-sets formed with cells of $\mathcal{C}$. We generalize the definition of neighborhood of $k$-sets of vertices from (Morris et al., 2019) to neighborhood of $k$-sets of cells.

**Definition B.7** (Neighborhood of $k$-sets of cells). Given a $k$-set of cells $s = \{\sigma_1, \ldots, \sigma_k\}$ in $[\mathcal{C}]^k$, we define its neighborhood as the function $\mathcal{N}_k : [\mathcal{C}]^k \mapsto \mathcal{P}([\mathcal{C}]^k)$ defined as:

$$\mathcal{N}_k(s) = \left\{ t \in [\mathcal{C}]^k \mid |s \cap t| = k - 1 \right\}. \tag{10}$$

**Definition B.8** (The CC $k$-Weisfeiler-Leman ($k$-CCWL) test on combinatorial complexes). Let $(\mathcal{C}, \ell)$ be a labeled combinatorial complex. Let $\mathcal{N}$ be a neighborhood on $\mathcal{C}$. The scheme proceeds as follows:

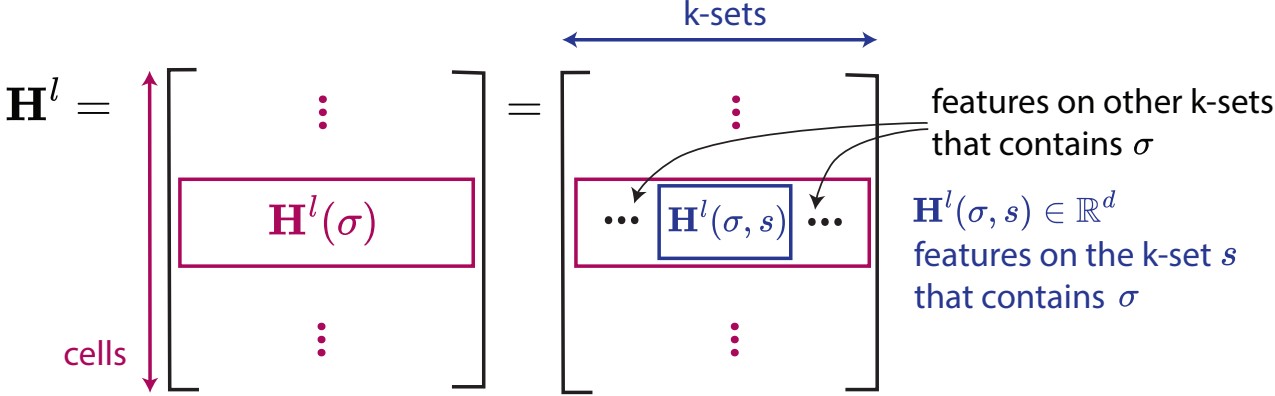

*Figure 8.* Notations for Proposition B.10. Denote $|\mathcal{C}|$ the number of cells. The number of $k$-sets that contain a given cell $\sigma$ is equal to $\binom{|\mathcal{C}|-1}{k-1}$. The feature on one $k$-set that contains a given cell has dimension $d$. Thus, $\mathbf{H}^l \in \mathbb{R}^{|\mathcal{C}| \times F}$ for $F = \binom{|\mathcal{C}|-1}{k-1}d$. We note that the $k$-sets for one row do not correspond to the $k$-sets of another row. However, for every row, there is the same number of $k$-sets that contain the cell $\sigma$ characteristic of that row.

- Initialization: Every $k$-set $s$ in $[\mathcal{C}]^k$ is initialized with a color that corresponds to the CC-isomorphism type of the sub-CC defined by $s = \{\sigma_1, \ldots, \sigma_k\}$ induced by $\mathcal{N}|_s$ where $\mathcal{N}|_s$ is the neighborhood $\mathcal{N}$ restricted to $s$. This means that two $k$-sets $s$ and $s'$ get the same color if and only if there is a labeled CC-isomorphism (for labeling function $\ell$) between the sub-CCs corresponding to the cells in $s$ and $s'$, respectively.

- Refinement: Given colors of $k$-sets at iteration $t$, the refinement step computes the color of the $k$-set $s$ at the next iteration $c_{s,\ell}^{t+1}$ using a perfect HASH function, as follows:

$$c_{\mathcal{N}_k(s),\ell}^t = \left\{\left\{c_{s',\ell}^t \mid \forall s' \in \mathcal{N}_k(s)\right\}\right\},$$
$$c_{s,\ell}^{t+1} = \mathrm{HASH}\left(c_{s,\ell}^t, c_{\mathcal{N}_k(s),\ell}^t\right).$$

- Termination: The algorithm stops when an iteration leaves the coloring unchanged.

Two combinatorial complexes are deemed non-isomorphic according to the CCWL and $k$-CCWL respectively, if their color histograms differ upon termination of the scheme. If the histograms are the same, we cannot conclude.

### B.3.3. DEFINITIONS OF $k$-GNNS AND $k$-CCNNS

We generalize the definition of $k$-GNNs by (Morris et al., 2019) into a definition of $k$-CCNNs.

**Definition B.9** ($k$-CCNNs). Let $(\mathcal{C}, \ell)$ be a labeled CC. In each $k$-CCNN layer $t$, the feature vector $h_k^{(t)}(s) \in \mathbb{R}^d$ for each $k$-set $s$ in $[\mathcal{C}]^k$ is updated into $h_k^{(t+1)}(s)$ as follows:

$$h_k^{(t+1)}(s) = U\left(h_k^{(t)}(s) \cdot W_1^{(t)} + \sum_{u \in \mathcal{N}_k(s)} h_k^{(t)}(u) \cdot W_2^{(t)}\right) \in \mathbb{R}^d, \tag{11}$$

where $W_1^{(t)}, W_2^{(t)}$ are matrices of parameters for layer $t$, $\mathcal{N}_k$ the neighborhood structure on $k$-sets, and $U$ is an update function.

Then, we show that $k$-CCNNs of Definition B.9 form a subclass of GCCNs.

**Proposition B.10.** *GCCNs generalize and subsume $k$-CCNNs.*

*Proof.* Let be given a $k$-CCNN defined by (11). We show that we can recover (11) by an appropriate choice of feature dimensionality $F$, update function $\phi$ and sub-module $\omega_{\mathcal{N}}$ in (8) defining GCCNs, and thus that any $k$-CCNN can be expressed as a GCCN.

For simplicity of notations, we assume that the layers of the $k$-CCNN have same feature dimensionality, denoted $d$. Given that there are $\binom{|\mathcal{C}|-1}{k-1}$ $k$-sets containing a given cell $\sigma$, we define $F = \binom{|\mathcal{C}|-1}{k-1}d$ to be the feature dimensionality of the layers of a GCCN. Denote $\mathbf{H}^l(\sigma)$ the row of $\mathbf{H}^l$ containing $F$-dimensional feature corresponding to cell $\sigma$, as well as $\mathbf{H}^l(\sigma, s)$ the subrow containing the $d$-dimensional feature corresponding to one $k$-set $s$ to which $\sigma$ belongs. Figure 8 illustrates these notations. We then define:

$$\mathbf{H}^{l+1} = \phi\left(\mathbf{H}^l, \omega(\mathbf{H}^l)\right)$$

by defining $\phi$ and $\omega$ on $(\sigma, s)$-blocks of the matrix $\mathbf{H}^l$. Specifically, we have:

$$\mathbf{H}^{l+1}(\sigma, s) = \phi_{(\sigma,s)}\left(\mathbf{H}^l(\sigma, s), \omega_{(\sigma,s)}(\mathbf{H}^l)\right)$$

$$= U\left(\mathbf{H}^l(\sigma, s) \cdot W_1^{(t)} + \sum_{u \in \mathcal{N}_k(s)} \mathbf{H}^l(\sigma, u) \cdot W_2^{(t)}\right),$$

where:

$$\omega_{(\sigma,s)}(\mathbf{H}^l) = \sum_{u \in \mathcal{N}_k(s)} \mathbf{H}^l(\sigma, u) \cdot W_2^{(t)}, \qquad \phi_{(\sigma,s)}(A, B) = U(A \cdot W_1^{(1)} + B). \tag{12}$$

In other words, we first use $\omega$ defined as a sequence of $\omega_{(\sigma,s)}$ to update each $(\sigma, s)$-block of $\mathbf{H}^l$ into an auxiliary feature $B = \tilde{\mathbf{H}}^l$. Then, we use $\phi$ as a sequence of $\phi_{(\sigma,s)}$ to perform a block-wise operations. Thus, we have built a GCCN that reproduces the computations of the $k$-CCNN. Therefore, GCCNs generalize and subsume $k$-CCNNs. $\square$

### B.3.4. RELATIONSHIPS BETWEEN CCWL/GCWL TESTS AND CCNNS/GCCNS

We prove relationships between the expressivity of the WL tests and the expressivity of the corresponding neural networks. We first recall results on WL tests on graphs and GNNs (Morris et al., 2019). In what follows, $(G, \ell)$ is a labeled graph, and $W^{(t)}$ denote the parameters of a GNN up to layer $t$. We encode the initial labels $\ell(v)$, for a vertex $v$, by vectors $h^{(0)}(v) \in \mathbb{R}^{1 \times d}$.

**WL/GNNs and $k$-WL/$k$-GNNs**  Theorem 1 in (Morris et al., 2019) states that, for every encoding of the graph labels $\ell(v)$ as $d$-vectors $h^{(0)}(v)$, and for every choice of parameters $W^{(t)}$, the coloring $c(t)_\ell$ of the WL test always refines the coloring $h(t)$ induced by the GNN parameterized by $W^{(t)}$. Theorem 2 in (Morris et al., 2019) states that there exists parameter matrices $W^{(t)}$ such that GNNs have exactly the same power as the WL test. Consequently, we say that GNNs have the same expressivity as the WL test. Similarly, Propositions 3 and 4 from (Morris et al., 2019) show that $k$-GNNs have the same expressivity as the k-WL test.

**CCWL/CCNNs and $k$-CCWL/GCCNs**  We generalize the equivalence between WL tests and GNNs to the framework of CCs. First, we prove two propositions establishing equivalence of WL tests between CCs and Hasse graphs.

**Proposition B.11** (CCWL and WL on the Hasse graph)**.** *Let $(\mathcal{C}, \ell)$ be a labeled CC. Let $\mathcal{N}$ be one neighborhood on this CC and $\mathcal{G}_{\mathcal{N}}$ the associated strictly augmented Hasse graph. The CCWL test defined in Def. B.6 is equivalent to the WL test defined on $\mathcal{G}_{\mathcal{N}}$.*

*Proof.* We prove the equivalence between the CCWL and the WL on $\mathcal{G}_{\mathcal{N}}$.

*Equivalence of initializations.* The CCWL test initializes cell colors using the labels given by $\ell$. The labeling function $\ell$ labels cells of $\mathcal{C}$ and therefore its restriction to $\mathcal{C}_{\mathcal{N}}$ labels nodes of the associated Hasse graph $\mathcal{G}_{\mathcal{N}}$. This turns $\mathcal{G}_{\mathcal{N}}$ into a labeled graph $(\mathcal{G}_{\mathcal{N}}, \ell_{\mathcal{C}_{\mathcal{N}}})$. We initialize the WL test on $\mathcal{G}_{\mathcal{N}}$ with colors from $\ell_{C_{\mathcal{N}}}$.

*Equivalence of refinements.* By construction of the strictly augmented Hasse graph $\mathcal{G}_{\mathcal{N}}$, nodes in $\mathcal{G}_{\mathcal{N}}$ are cells in $\mathcal{C}_{\mathcal{N}}$ and edges in $\mathcal{G}_{\mathcal{N}}$ are neighbors in $\mathcal{C}_{\mathcal{N}}$ for the neighborhood $\mathcal{N}$. Thus, the refinement equation of the CCWL test is equal to the refinement equation of the WL test on $\mathcal{G}_{\mathcal{N}}$. This proves that CCWL and the WL on $\mathcal{G}_{\mathcal{N}}$ are equivalent. $\square$

**Proposition B.12** ($k$-CCWL and $k$-WL on the Hasse graph)**.** *Let* $(\mathcal{C}, \ell)$ *be a labeled CC. Let* $\mathcal{N}$ *be one neighborhood on this CC and* $\mathcal{G}_{\mathcal{N}}$ *the associated strictly augmented Hasse graph. The* $k$-*CCWL defined in Def. B.8 is equivalent to the* $k$-*WL test on* $\mathcal{G}_{\mathcal{N}}$.

*Proof.* We prove the equivalence between the $k$-CCWL and the $k$-WL on $\mathcal{G}_{\mathcal{N}}$.

*Equivalence of initializations.* The $k$-CCWL test initializes colors of $k$-sets based on the CC-isomorphism class of every sub-CC defined by every $k$-set. Using Proposition B.4, the CC-isomorphism class of a sub-CC $s$ corresponds to the graph isomorphism class on the associated subgraph in the strictly augmented Hasse graph. We initialize the $k$-WL test on $\mathcal{G}_{\mathcal{N}}$ with colors on $k$-sets associated with this isomorphism class.

*Equivalence of refinements.* By construction of the strictly augmented Hasse graph $\mathcal{G}_{\mathcal{N}}$, $k$-sets of nodes in $\mathcal{G}_{\mathcal{N}}$ are $k$-sets of cells in $\mathcal{C}_{\mathcal{N}}$, and the neighborhoods of $k$-sets of nodes defined in (Morris et al., 2019) are the neighborhoods of $k$-sets of cells defined in Definition B.7. Thus, the refinement equation of the $k$-CCWL test is equal to the refinement equation of the $k$-WL test on $\mathcal{G}_{\mathcal{N}}$.

This proves that $k$-CCWL and the $k$-WL on $\mathcal{G}_{\mathcal{N}}$ are equivalent. □

Given the equivalence between the computations in $\mathcal{C}$ and in $\mathcal{G}_{\mathcal{N}}$ provided by Proposition B.11, we can pull the results from Theorems 1 and 2 from (Morris et al., 2019) and provide the following propositions.

**Proposition B.13.** *Let* $(\mathcal{C}, \ell)$ *be a labeled CC. Then for all* $t \geq 0$ *and for all choices of initial colorings* $h^{(0)}$ *consistent with* $\ell$, *and weights* $\mathbf{W}^{(t)}$, $c_{\ell}^{(t)} \sqsubseteq h^{(t)}$, *i.e., the coloring* $c_l^{(t)}$ *induced by the CCWL test refines the coloring induced by the CCNN* $h^{(t)}$.

**Proposition B.14.** *Let* $(\mathcal{C}, \ell)$ *be a labeled CC. Then for all* $t \geq 0$ *there exists a sequence of weights* $\mathbf{W}^{(t)}$, *and a CCNN architecture such that* $c_{\ell}^{(t)} \equiv h^{(t)}$., *i.e., the coloring of the CCWL and the CCNN are equivalent.*

Consequently, CCNNs have the same power as the CCWL. Next, we measure the power of $k$-CCNNs using the $k$-CCWL.

**Proposition B.15.** *Let* $(\mathcal{C}, \ell)$ *be a labeled CC and let* $k \geq 2$. *Then, for all* $t \geq 0$, *for all choices of initial colorings* $h_k^{(0)}$ *consistent with* $\ell$ *and for all weights* $\mathbf{W}^{(t)}$, $c_{s,k,\ell}^{(t)} \sqsubseteq h_k^{(t)}$ *i.e., the coloring* $c_{s,k,l}^{(t)}$ *induced by the* $k$-*CCWL test refines the coloring induced by the* $k$-*CCNN* $h_k^{(t)}$.

**Proposition B.16.** *Let* $(\mathcal{C}, \ell)$ *be a labeled CC and let* $k \geq 2$. *Then, for all* $t \geq 0$ *there exists a sequence of weights* $\mathbf{W}^{(t)}$, *and a* $k$-*CCNN architecture such that* $c_{s,k,\ell}^{(t)} \equiv h_k^{(t)}$.

Propositions B.15 and B.16 are given by Proposition B.12 and Propositions 3 and 4 from Morris et al. (2019). They show that $k$-CCNNs have the same power as the $k$-CCWL.

B.3.5. PROOF

We now provide the proof for Proposition 4.3 that states that GCCNs are strictly more expressive than CCNNs.

*Proof.* We prove that GCCNs are strictly more powerful than CCNNs in distinguishing non-isomorphic combinatorial complexes. We leverage the propositions of this subsection summarized on Figure 7.

By Proposition B.10, GCCN have at least the same expressive power as $k$-CCNNs. By Propositions B.15-B.16, $k$-CCNNs have the same expressive power as the $k$-CCWL. By Proposition B.12, the $k$-CCWL test is equivalent to the $k$-WL test on the associated strictly augmented Hasse graph. It is known (e.g., (Grohe, 2017)) that the $k$-WL test on graph is strictly more powerful than the WL test. Thus, the $k$-WL test on the strictly augmented Hasse graph is strictly more powerful than the WL test on that same graph. By Proposition B.11, the WL test on the strictly augmented Hasse graph is equivalent to the CCWL test on the corresponding CC. By Propositions B.13-B.14, the CCWL test on the CC has the same expressive power as CCNNs.

Consequently, we have shown that GCCN are strictly more powerful than CCNNs in distinguishing nonisomorphic CCs. □

Additionally, we construct two combinatorial complexes $\mathcal{C}_1$ and $\mathcal{C}_2$ that are indistinguishable by CCNNs but distinguishable by GCCNs.

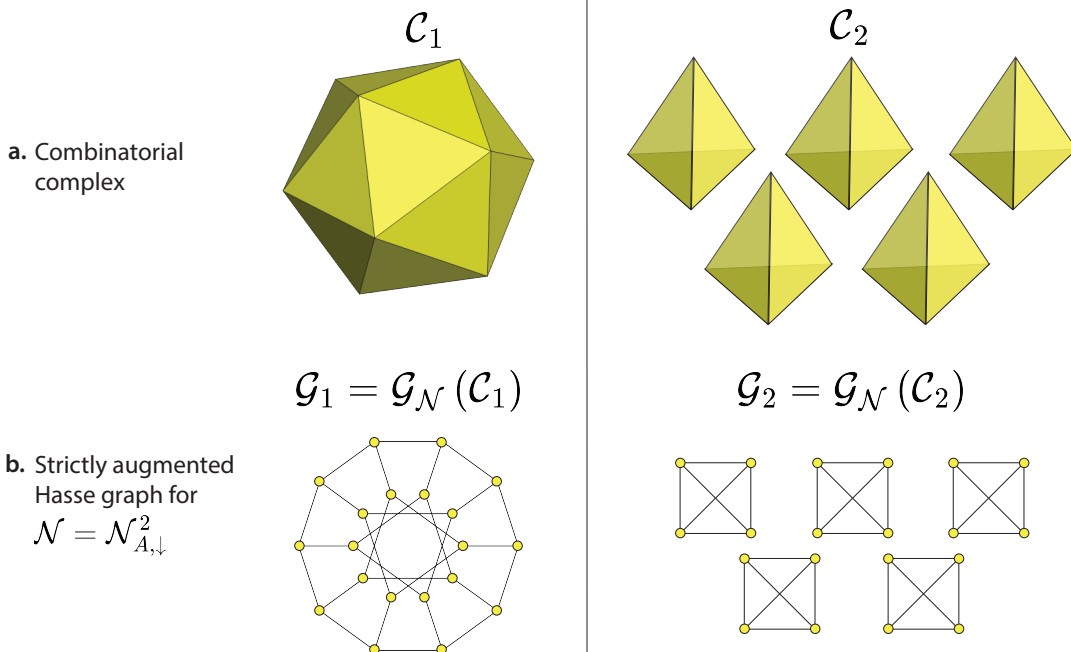

*Figure 9.* **a.** Pair of combinatorial complexes: $\mathcal{C}_1$ is an icosahedron polygon, and $\mathcal{C}_2$ is five tetrahedrons. **b.** Strictly augmented Hasse graphs corresponding to each combinatorial complex, given a choice of neighborhood $\mathcal{N}_{A,\uparrow}^2$.

Let $\mathcal{C}_1$ and $\mathcal{C}_2$ be two combinatorial complexes with a neighborhood structure $\mathcal{N}_{\mathcal{C}} = \mathcal{N}_{A,\downarrow}^2$ (down-adjacency of faces). These complexes are illustrated in Figure 9a.

The corresponding strictly augmented Hasse graphs $\mathcal{G}_1$ and $\mathcal{G}_2$ (Fig. 9b) represent the 20 faces of each complex as nodes, where each node has degree 3. Thus:

- Both $\mathcal{G}_1$ and $\mathcal{G}_2$ are 3-regular graphs.

- It is known that regular graphs of the same order are indistinguishable by the WL test (see, e.g., (Kiefer, 2020; Morris et al., 2023)).

- Every pair of graphs with $n$ nodes are distinguishable by the $n$-WL test (Morris et al., 2023).

Since CCWL is equivalent to WL on $\mathcal{G}_{\mathcal{N}}$ (Proposition B.11), the two complexes $\mathcal{C}_1$ and $\mathcal{C}_2$ are indistinguishable by CCWL. Since CCWL has the same expressive power as CCNNs (Propositions B.13-B.14), the two complexes $\mathcal{C}_1$ and $\mathcal{C}_2$ are indistinguishable by CCNNs.

Since $k$-CCWL is equivalent to $k$-WL on $\mathcal{G}_{\mathcal{N}}$ (Proposition B.12), the two complexes $\mathcal{C}_1$ and $\mathcal{C}_2$ are distinguishable by $k$-CCWL. Since $k$-CCWL has the same expressive power as $k$-CCNNs (Propositions B.15-B.16), the two complexes $\mathcal{C}_1$ and $\mathcal{C}_2$ are distinguishable by $k$-CCNNs. Since GCCNs generalize and subsume $k$-CCNNs (Proposition B.10), $\mathcal{C}_1$ and $\mathcal{C}_2$ are distinguishable by GCCNs.

Thus, we have constructed two combinatorial complexes $\mathcal{C}_1$ and $\mathcal{C}_2$ that are indistinguishable by CCNNs, but are distinguishable by GCCNs.

## C. Time Complexity

To analyze the time complexity (in terms of FLOPs) of the Generalized Combinatorial Complex Neural Network (GCCN), we derive the complexity of its submodule $\omega_{\mathcal{N}}$ and then compute the complexity of a GCCN layer. We then compare it with GNN and CCNN complexity.

### C.1. Key Definitions

- **Message Complexity** ($M$)**:** The complexity of a single message computation along a route (e.g., node $\to$ node). For example, in a Graph Convolutional Network (GCN), a single message is defined as:

$$m_{x \to y} = a_{xy} \mathbf{h}_y \Theta,$$

where $\mathbf{h}_y$ is a $1 \times F$ vector, $\Theta$ is an $F \times F$ weight matrix, and $a_{xy}$ is a scalar. This involves a matrix-vector multiplication, contributing a complexity of $O(F^2)$ per message.

- **Update Complexity** ($U$)**:** The complexity of the update function in the reference GNN. For simplicity, we assume the update is an element-wise function, giving $U = O(|N|)$, where $|N|$ is the number of nodes.

### C.2. Complexity of $\omega_{\mathcal{N}}$

Assuming each $\omega_{\mathcal{N}}$ submodule is a single-layer GNN, the complexity of $\omega_{\mathcal{N}}$ can be decomposed into three components: **message computation, aggregation, and update.**

$$C_{\omega_{\mathcal{N}}} = C_{\text{message}} + C_{\text{aggregation}} + C_{\text{update}}$$

This breaks down as:

$$C_{\omega_{\mathcal{N}}} = 2|E|M + \sum_{n \in N} \deg(n)A + |N|U,$$

where:

- $|E|$: Number of edges in the graph,

- $M$: Complexity per message ($O(F^2)$),

- $\deg(n)$: Degree of node $n$,

- $A$: Complexity of aggregation (e.g., assuming sum/average, $O(F)$),

- $U$: Complexity of the update function ($O(1)$ per node).

Substituting assumptions for convolutional message passing, summation aggregation, and constant node degree $d$:

$$C_{\omega_{\mathcal{N}}} = 2|E|F^2 + \sum_{n \in N} \deg(n)F + O(|N|),$$

$$C_{\omega_{\mathcal{N}}} = 2|E|F^2 + |N|dF + O(|N|),$$
$$C_{\omega_{\mathcal{N}}} = O(|E|F^2 + |N|dF + |N|).$$

### C.3. Complexity Using Combinatorial Complex Notations

Up until now, we have expressed $C_{\omega_{\mathcal{N}}}$ in terms of the nodes and edges making up the strictly expanded Hassse graph it receives as input. To be able to write the complexity of a whole GCCN layer, we must express $C_{\omega_{\mathcal{N}}}$ in terms of the original cells represented as nodes in the graph. Specifically, we will denote the source cells (cells sending messages) as cells of rank $r$ and the destination cells (cells receiving messages) as cells of rank $r'$. The relationships governing adjacency between the nodes representing these cells will come from the neighborhood $\mathcal{N}$ to which the submodule $\omega_{\mathcal{N}}$ is assigned.

Rewriting in terms of combinatorial complex notations, where:

- $\|\mathcal{N}\|_0$: Total number of relationships in $\mathcal{N}$ (i.e. number of nonzero entries in matrix corresponding to $\mathcal{N}$),

- $n_{r'}$: Number of $r'$-cells.

- $d_{r'}$: Assumed constant degree of $r'$-cells,

The complexity becomes:

$$C_{\omega_{\mathcal{N}}} = O(\|\mathcal{N}\|_0 F^2 + \text{nrows}(\mathcal{N})d_{r'}F + n_{r'}),$$
$$C_{\omega_{\mathcal{N}}} = O(\|\mathcal{N}\|_0 F^2 + \text{nrows}(\mathcal{N})d_{r'}F + \text{nrows}(\mathcal{N})).$$

### C.4. Complexity of a GCCN Layer

A GCCN layer is composed of a set of $\omega_{\mathcal{N}}$'s, one for each $\mathcal{N} \in \mathcal{N}_{\mathcal{C}}$. The complexity of a GCCN layer is the sum of all the complexities of its submodules, plus the complexity of the module responsible for aggregating the outputs of each neighborhood, i.e. the inter-neighborhood aggregation. We assume this inter-aggregation to be a sum. The layer complexity is:

$$C_{\text{GCCN}} = \sum_{\mathcal{N} \in \mathcal{N}_{\mathcal{C}}} C_{\omega_{\mathcal{N}}} + C_{\text{inter-agg}},$$

where:

$$C_{\text{inter-agg}} = \sum_{r' \in [0, R']} n_{r'} n_{\mathcal{N}_{r'}} F,$$

and $n_{\mathcal{N}_{r'}}$ is the number of neighborhoods sending messages to $r'$-cells.

### C.5. Takeaways

- **GNN Comparison:** GCCNs increase complexity compared to traditional GNNs due to :
  - the introduction of multiple neighborhoods. A GCCN considers many $\mathcal{N} \in \mathcal{N}_{\mathcal{C}}$, going beyond the simple node-level adjacency $\mathcal{N}_{\mathcal{C}} = A_0$ of a GNN. This is what allows TDL models (GCCNs and CCNNs) to operate on a richer topological space than GNNs.
  - inter-neighborhood aggregation.

- **CCNN Comparison:** Unlike traditional CCNNs, GCCNs allow per-rank neighborhoods, enabling many smaller possible sets of neighborhoods $\mathcal{N}_{\mathcal{C}}$. This more selective inclusion of neighborhoods reduces redundancy. Concretely, this means the sum $\sum_{\mathcal{N} \in \mathcal{N}_{\mathcal{C}}} C_{\omega_{\mathcal{N}}}$ can be smaller.

- **Tradeoff:** GCCNs' time complexity are a compromise between GNNs and CCNNs. While they do introduce $C_{\text{inter-agg}}$ (like CCNNs) and additional elements to the sum $\sum_{\mathcal{N} \in \mathcal{N}_{\mathcal{C}}} C_{\omega_{\mathcal{N}}}$, they can introduce less elements to this sum than CCNNs.

## D. Software

Algorithm 1 shows how the TopoTune module instantiates a GCCN by taking a choice of model $\omega_{\mathcal{N}}$ and neighborhoods $\mathcal{N}_{\mathcal{C}}$ as input. Given an input complex $x$, TopoTune first expands it into an ensemble of strictly augmented Hasse graphs that are then passed to their respective $\omega_{\mathcal{N}}$ models within each GCCN layer.

*Remark.* We decided to design the software module of TopoTune, i.e., how to implement GCCNs, as we did for mainly two reasons: (i) the full compatibility with TopoBench (implying consistency of the combinatorial complex instantiations and the benchmarking pipeline), and (ii) the possibility of using GNNs as neighborhood message functions that are not necessarily implemented with a specific library. However, if the practitioner is interested in entirely wrapping the GCCN implementation into Pytorch Geometric or DGL, they can do it by noticing that a GCCN is equivalent to a *heterogeneous* GNN where the heterogeneous graph the whole augmented Hasse graph, with node types given by the rank of the cell (e.g. 0-cells, 1-cells, and 2-cells) while the edge type is given by the per-rank neighborhood function (e.g. "0-cells to 1-cells" or "2-cells to 1-cells" for $\mathcal{N}_{I,\uparrow}^0$ and $\mathcal{N}_{I,\downarrow}^2$, respectively).

---

**Algorithm 1** TopoTune

    **Class** TopoTune(torch.nn.Module):
0: **procedure** INIT(neighborhoods, $\omega_n$, $\omega_n\_params$, layers)
0:    $self.omega\_n\_submodels \leftarrow []$
0:    **for** $l \leftarrow 1$ **to** layers **do**
0:        $layer\_models \leftarrow []$
0:        **for** each $nb$ in neighborhoods **do**
0:            $model \leftarrow \omega_n(\omega_n\_params)$
0:            $layer\_models.append(model)$
0:        **end for**
0:        $self.omega\_n\_submodels.append(layer\_models)$
0:    **end for**
0: **end procedure**
0: **procedure** FORWARD($x$)
0:    **for** each layer in $self.omega\_n\_submodels$ **do**
0:        $outputs \leftarrow []$
0:        **for** each $\omega_n\_model$ in layer **do**
0:            $hasse\_graph \leftarrow self.expand\_to\_strictly\_aug\_hasse\_graph(x)$
0:            $outputs.append(\omega_n\_model(hasse\_graph))$
0:        **end for**
0:        $x \leftarrow self.aggregate\_rank\_wise(outputs)$
0:    **end for**
0:    **return** $x$
0: **end procedure**

    **Example Instantiation:**
0: $neighborhoods \leftarrow [[[0,0], up\_adjacency], [[2,1], incidence]]$
0: $\omega_n \leftarrow$ torch_geometric.nn.models.GAT
0: $\omega_n\_params \leftarrow \{num\_layers : 2, heads : 4\}$
0: $layers \leftarrow 4$
0: $model \leftarrow$ TopoTune$(neighborhoods, \omega_n, \omega_n\_params, layers)$
    =0

---

# E. Additional details on experiments

In this section, we delve into the details of the datasets, hyperparameter search methodology, and computational resources utilized for conducting the experiments.

## E.1. Neighborhood Structures

In order to build a broad class of GCCNs, we consider X different neighborhood structures on which we perform graph expansion. Importantly, three of these structures are lightweight, per-rank neighborhood structures, as proposed in Section 4. The neighborhood structures are:

$$
\begin{aligned}
& \left\{\mathcal{N}_{A,\uparrow}^{0}, \mathcal{N}_{A,\uparrow}^{1}\right\} \quad \left\{\mathcal{N}_{A,\uparrow}^{0}, \mathcal{N}_{I,\downarrow}^{2}\right\} \quad \{\mathcal{N}_{A,\uparrow}, \mathcal{N}_{I,\uparrow}\} \quad \{\mathcal{N}_{A,\uparrow}, \mathcal{N}_{A,\downarrow}, \mathcal{N}_{I,\downarrow}\} \quad \{\mathcal{N}_{A,\uparrow}\} \\
& \left\{\mathcal{N}_{A,\uparrow}, \mathcal{N}_{A,\downarrow}^{1}\right\} \quad \{\mathcal{N}_{A,\uparrow}, \mathcal{N}_{A,\downarrow}\} \quad \{\mathcal{N}_{A,\uparrow}, \mathcal{N}_{I,\downarrow}\} \quad \{\mathcal{N}_{A,\uparrow}, \mathcal{N}_{A,\downarrow}, \mathcal{N}_{I,\uparrow}\} \quad \{\mathcal{N}_{A,\uparrow}, \mathcal{N}_{A,\downarrow}, \mathcal{N}_{I,\downarrow}, \mathcal{N}_{I,\uparrow}\}
\end{aligned}
$$

## E.2. Datasets

### Dataset statistics

Table 3 provides the statistics for each dataset lifted to three topological domains: simplicial complex, cellular complex, and hypergraph. The table shows the number of 0-cells (nodes), 1-cells (edges), and 2-cells (faces) of each dataset after the topology lifting procedure. We recall that:

- the simplicial clique complex lifting is applied to lift the graph to a simplicial domain, with a maximum complex dimension equal to 2;

- the cellular cycle-based lifting is employed to lift the graph into the cellular domain, with maximum complex dimension set to 2 as well.

*Table 3.* Descriptive summaries of the datasets used in the experiments.

| Dataset | Domain | # 0-cell | # 1-cell | # 2-cell |
|---|---|---|---|---|
| Cora | Cellular | 2,708 | 5,278 | 2,648 |
| | Simplicial | 2,708 | 5,278 | 1,630 |
| Citeseer | Cellular | 3,327 | 4,552 | 1,663 |
| | Simplicial | 3,327 | 4,552 | 1,167 |
| PubMed | Cellular | 19,717 | 44,324 | 23,605 |
| | Simplicial | 19,717 | 44,324 | 12,520 |
| MUTAG | Cellular | 3,371 | 3,721 | 538 |
| | Simplicial | 3,371 | 3,721 | 0 |
| NCI1 | Cellular | 122,747 | 132,753 | 14,885 |
| | Simplicial | 122,747 | 132,753 | 186 |
| NCI109 | Cellular | 122,494 | 132,604 | 15,042 |
| | Simplicial | 122,494 | 132,604 | 183 |
| PROTEINS | Cellular | 43,471 | 81,044 | 38,773 |
| | Simplicial | 43,471 | 81,044 | 30,501 |
| ZINC (subset) | Cellular | 277,864 | 298,985 | 33,121 |
| | Simplicial | 277,864 | 298,985 | 769 |

**Dataset selection and limitations**

The datasets employed in this work and other TDL studies are predominantly adapted from the GNN literature. Among these, molecular datasets stand out due to the inherent importance of cycles and hyperedges, which effectively capture chemical rings and functional groups. These are structures that are naturally represented in topological domains.

While TDL methods are not intrinsically constrained to these datasets, the lifting procedures used to construct higher-order cells introduce computational bottlenecks, particularly in memory usage. For instance, operations such as cycle detection and clique enumeration, required for constructing cellular complexes or simplicial complexes, respectively, become computationally prohibitive for large or densely connected graphs.

To address these limitations, ongoing research is focused on developing scalable lifting procedures that can extend TDL methods to broader datasets, including those with more complex structures or larger scales. For example, Bernárdez et al. (2024) propose innovative topological liftings, paving the way for more scalable and applicable datasets in TDL.

## E.3. Hyperparameter search

Five splits are generated for each dataset to ensure a fair evaluation of the models across domains. Each split comprises 50% training data, 25% validation data, and 25% test data. An exception is made for the ZINC dataset, where predefined splits are used (Irwin et al., 2012).

To avoid the combinatorial explosion of possible hyperparameter sets, we fix the values of all hyperparameters beyond GCCNs: hence, to name a few relevant parameters, we set the learning rate to $0.01$, the batch size to the default value of TopoBench for each dataset, and the cell hidden state dimension to 32. Regarding the internal GCCN hyperparameters, a grid-search strategy is employed to find the optimal set for each model and dataset. Specifically, we consider 10 different neighborhood structures (see Section E.1), and the number of GCCN layers is varied over $\{2, 4, 8\}$. For GNN-based neighborhood message functions, we vary over {GCN,GAT,GIN,GraphSage} models from PyTorch Geometric, and for each of them consider either 1 or 2 number of layers. For the Transformer-based neighborhood message function (Transformer Encoder model from PyTorch), we vary the number of heads over $\{2, 4\}$, and the feed-forward neural network dimension over $\{64, 128\}$.

For node-level task datasets, validation is conducted after each training epoch, continuing until either the maximum number of epochs is reached or the optimization metric fails to improve for 50 consecutive validation epochs. The minimum number of epochs is set to 50. Conversely, for graph-level tasks, validation is performed every 5 training epochs, with training halting if the performance metric does not improve on the validation set for the last 10 validation epochs. To optimize the models, `torch.optim.Adam` is combined with `torch.optim.lr_scheduler.StepLR` wherein the step size was set to 50 and the gamma value to 0.5. The optimal hyperparameter set is generally selected based on the best average performance over five validation splits. For the ZINC dataset, five different initialization seeds are used to obtain the average performance.

## E.4. Hardware

The hyperparameter search is executed on a Linux machine with 256 cores, 1TB of system memory, and 8 NVIDIA A100 GPUs, each with 80GB of GPU memory.

# F. Model Size

We provide details on model size for reported results in Section 6.

*Table 4.* Model size corresponding to results reported in Table 1.

| | Graph-Level Tasks | | | | | Node-Level Tasks | | |
|---|---|---|---|---|---|---|---|---|
| Model | MUTAG | PROTEINS | NCI1 | NCI109 | ZINC | Cora | Citeseer | PubMed |
| **Cellular** | | | | | | | | |
| CCNN (Best Model on TopoBench) | 334.72K | 101.12K | 63.87K | 17.67K | 88.06K | 451.85K | 1032.84K | 163.72K |
| GCCN $\omega_{\mathcal{N}}$ = GAT | 15.11K | 46.27K | 68.99K | 49.63K | 39.78K | 341.54K | 1677.32K | 344.83K |
| GCCN $\omega_{\mathcal{N}}$ = GCN | 45.44K | 45.25K | 65.92K | 30.69K | 29.54K | 801.16K | 1507.59K | 443.91K |
| GCCN $\omega_{\mathcal{N}}$ = GIN | **63.62K** | 23.49K | 49.03K | **66.79K** | **64.35K** | 669.58K | 1674.25K | 211.97K |
| GCCN $\omega_{\mathcal{N}}$ = GraphSAGE | 44.42K | 76.99K | **47.49K** | 115.17K | 79.71K | 1195.14K | 741.5K | 640.51K |
| GCCN $\omega_{\mathcal{N}}$ = Transformer | 112.26K | 78.79K | 82.05K | 115.43K | 317.02K | 249.51K | 468.29K | 331.59K |
| GCCN $\omega_{\mathcal{N}}$ = Best GNN, 1 Hasse graph | 14.98K | 18.88K | 18.05K | 15.91K | 20.83K | 150.12K | 367.88K | 66.50K |
| **Simplicial** | | | | | | | | |
| CCNN (Best Model on TopoBench) | 398.85K | 10.24K | 131.84K | 135.75K | 617.86K | 144.62K | 737.29K | 134.40K |
| GCCN $\omega_{\mathcal{N}}$ = GAT | 15.11K | 46.27K | 68.99K | 49.63K | 67.42K | 341.45K | 1677.32K | 344.83K |
| GCCN $\omega_{\mathcal{N}}$ = GCN | 45.44K | 45.25K | 65.92K | 30.69K | 64.35K | 801.16K | 1507.59K | 443.91K |
| GCCN $\omega_{\mathcal{N}}$ = GIN | 63.62K | 23.49K | 49.03K | 66.79K | 118.11K | 669.58K | 1674.25K | 211.97K |
| GCCN $\omega_{\mathcal{N}}$ = GraphSAGE | 44.42K | 76.99K | 47.49K | 115.17K | 147.30K | 1195.14K | **741.51K** | 640.51K |
| GCCN $\omega_{\mathcal{N}}$ = Transformer | 113.15K | 213.70K | 82.05K | 166.24K | 148.83K | 284.58K | 468.29K | 331.59K |
| GCCN $\omega_{\mathcal{N}}$ = Best GNN, 1 Hasse graph | 19.07K | 14.66K | 31.11K | 15.91K | 29.54K | 150.12K | 367.88K | 66.50K |
| **Hypergraph** | | | | | | | | |
| CCNN (Best Model on TopoBench) | 84.10K | **14.34K** | 88.19K | 88.32K | 22.53K | **60.26K** | 258.50K | **280.83K** |

*Table 5.* Model sizes corresponding to results in Table 2.

| | Graph-Level Tasks | | | | Node-Level Tasks | | |
|---|---|---|---|---|---|---|---|
| Model | MUTAG | PROTEINS | NCI1 | NCI109 | Cora | Citeseer | PubMed |
| **SCCN** | | | | | | | |
| TopoBench | 398.85K | 397.31K | **131.84K** | **135.75K** | 155.88K | 782.34K | **457.99K** |
| 1 Hasse graph / $\mathcal{N}$, $\omega_{\mathcal{N}}$ = Best(GNN) | **852.74K** | **851.97K** | 248.58K | 291.39K | **159.46K** | 791.56K | 510.47K |
| 1 Hasse graph for $\{\mathcal{N}\}$, $\omega_{\mathcal{N}}$ = Best(GNN) | 104.32K | 153.09K | 71.17K | 54.85K | 143.66K | **741.51K** | 376.58K |
| **CWN** | | | | | | | |
| TopoBench | 334.72K | **101.12K** | 124.10K | 412.29K | 343.11K | **1754.50K** | **163.72K** |
| 1 Hasse graph / $\mathcal{N}$, $\omega_{\mathcal{N}}$ = Best(GNN) | **350.46K** | 353.54K | 95.75K | 465.28K | 900.23K | 177.10K | 159.56K |
| 1 Hasse graph for $\{\mathcal{N}\}$, $\omega_{\mathcal{N}}$ = Best(GNN) | 219.65K | 283.91K | **78.85K** | **264.45K** | **138.95K** | 163.94K | 138.95K |

# G. Model Training Time

We provide training times for all experiments reported on in Section 6. We measure these training times by running each experiment on a single A30 NVIDIA GPU. We note that these times include the on-the-fly graph expansion method, which slows down the model forward proportionally to dataset size. We plan on moving this process into data preprocessing in the future.

*Table 6.* Model training time (seconds) corresponding to results reported in Table 1.

| Model | Graph-Level Tasks | | | | | Node-Level Tasks | | |
|---|---|---|---|---|---|---|---|---|
| | MUTAG (↑) | PROTEINS (↑) | NCI1 (↑) | NCI109 (↑) | ZINC (↓) | Cora (↑) | Citeseer (↑) | PubMed (↑) |
| **Cellular** | | | | | | | | |
| CCNN (Best Model on TopoBench) | 100 ± 23 | 132 ± 19 | 238 ± 89 | 254 ± 39 | 228 ± 44 | 75 ± 15 | 57 ± 4.4 | 128 ± 50 |
| GCCN $\omega_\mathcal{N}$ = GAT | 80 ± 11 | 64 ± 10 | 778 ± 118 | 486 ± 75 | 3173 ± 954 | 46 ± 3 | 63 ± 1 | 202 ± 22 |
| GCCN $\omega_\mathcal{N}$ = GCN | 43 ± 7 | 67 ± 16 | 544 ± 40 | 495 ± 108 | 4013 ± 620 | 46 ± 4 | 65 ± 3 | 149 ± 12 |
| GCCN $\omega_\mathcal{N}$ = GIN | **61 ± 18** | 59 ± 18 | 523 ± 119 | **386 ± 76** | **3301 ± 440** | 64 ± 8 | 77 ± 2 | 207 ± 33 |
| GCCN $\omega_\mathcal{N}$ = GraphSAGE | 43 ± 12 | 43 ± 3 | **691 ± 80** | 364 ± 102 | 2863 ± 262 | 49 ± 2 | 60 ± 3 | 211 ± 25 |
| GCCN $\omega_\mathcal{N}$ = Transformer | 50 ± 19 | 786 ± 147 | 1005 ± 27 | 1484 ± 181 | 15320 ± 5386 | 121 ± 20 | 94 ± 20 | 5459 ± 1374 |
| GCCN $\omega_\mathcal{N}$ = Best GNN, 1 Aug. Hasse graph | 33 ± 7 | 70 ± 24 | 451 ± 123 | 441 ± 130 | 3162 ± 340 | 47 ± 5 | 72 ± 6 | 194 ± 35 |
| **Simplicial** | | | | | | | | |
| CCNN (Best Model on TopoBench) | 123 ± 57 | 104 ± 28 | 172 ± 50 | 183 ± 62 | 178 ± 86 | 143 ± 16 | 75 ± 23 | 114 ± 18 |
| GCCN $\omega_\mathcal{N}$ = GAT | 25 ± 5 | 70 ± 17 | 755 ± 158 | 794 ± 151 | 2242 ± 275 | 49 ± 3 | 68 ± 2 | 192 ± 38 |
| GCCN $\omega_\mathcal{N}$ = GCN | 40 ± 7 | 138 ± 26 | 548 ± 185 | 603 ± 181 | 2428 ± 833 | 49 ± 5 | 67 ± 2 | 167 ± 22 |
| GCCN $\omega_\mathcal{N}$ = GIN | 61 ± 7 | 66 ± 21 | 904 ± 180 | 538 ± 39 | 3603 ± 475 | 71 ± 6 | 77 ± 8 | 210 ± 42 |
| GCCN $\omega_\mathcal{N}$ = GraphSAGE | 31 ± 3 | 61 ± 27 | 572 ± 124 | 511 ± 74 | 1721 ± 201 | 51 ± 3 | 74 ± 8 | 221 ± 37 |
| GCCN $\omega_\mathcal{N}$ = Transformer | 35 ± 5 | 947 ± 333 | 1386 ± 404 | 1360 ± 410 | 7979 ± 1373 | 146 ± 58 | 77 ± 2 | 5281 ± 827 |
| GCCN $\omega_\mathcal{N}$ = Best GNN, 1 Aug. Hasse graph | 25 ± 2 | 78 ± 27 | 598 ± 31 | 312 ± 7 | 2681 ± 910 | 52 ± 4 | 72 ± 8 | 156 ± 16 |
| **Hypergraph** | | | | | | | | |
| CCNN (Best Model on TopoBench) | 127 ± 48 | **96 ± 20** | 220 ± 74 | 128 ± 49 | 387 ± 105 | 121 ± 38 | 48 ± 1 | 177 ± 71 |

*Table 7.* Model training times (seconds) corresponding to results in Table 2.

| Model | Graph-Level Tasks | | | | Node-Level Tasks | | |
|---|---|---|---|---|---|---|---|
| | MUTAG | PROTEINS | NCI1 | NCI109 | Cora | Citeseer | PubMed |
| **SCCN (Yang et al., 2022)** | | | | | | | |
| **Benchmark results (Telyatnikov et al., 2024)** | 11 ± 2 | 60 ± 18 | **247 ± 65** | **311 ± 83** | 102 ± 39 | 101 ± 41 | **143 ± 35** |
| GCCN, on ensemble of strictly aug. Hasse graphs *2, dig | **14 ± 1** | **75 ± 8** | 413 ± 120 | 298 ± 15 | **121 ± 2** | 172 ± 6 | 285 ± 20 |
| GCCN, on 1 aug. Hasse graph *2, dig | 5 ± 1 | 59 ± 10 | 283 ± 90 | 217 ± 100 | 110 ± 3 | **166 ± 10** | 376 ± 27 |
| **CWN (Bodnar et al., 2021a)** | | | | | | | |
| **Benchmark results (Telyatnikov et al., 2024)** | 11 ± 2 | **43 ± 5** | 240 ± 50 | 252 ± 92 | 54 ± 25 | **52 ± 5** | **119 ± 14** |
| GCCN, on ensemble of strictly aug. Hasse graphs *2, dig | **12 ± 1** | 73 ± 10 | 536 ± 38 | 426 ± 90 | 91 ± 17 | 49 ± 1 | 125 ± 19 |
| GCCN, on 1 aug. Hasse graph *2, dig | 11 ± 1 | 62 ± 11 | **573 ± 107** | **410 ± 64** | **96 ± 2** | 46 ± 1 | 130 ± 20 |

# H. Additional experiments

## H.1. Larger node-level datasets

Table 8 additionally presents the experimental results on 4 heterophilic datasets introduced in (Platonov et al.) (Amazon Ratings, Roman Empire, Minesweeper, and Questions). These represent larger node-level classification tasks than those shown in the main Table 1, with up to 48,921 nodes and 153,540 edges in the case of the Questions graph. Except on this precise dataset, which was not considered in previous TDL literature, we compare the results against CCNNs and hypergraph models from Telyatnikov et al. (2024). We observe that overall GCCNs achieve similar performance than regular CCNNs, and they outperform them by a significant margin on Minesweeper.

|  | **Amazon Ratings** | **Roman Empire** | **Minesweeper** | **Questions** |
|---|---|---|---|---|
| Best GCCN Cell | 50.17 ± 0.71 | 84.48 ± 0.29 | 94.02 ± 0.28 | 78.04 ± 1.34 |
| Best CCNN Cell | **51.90 ± 0.15** | 82.14 ± 0.00 | 89.42 ± 0.00 | - |
| Best GCCN Simplicial | 50.53 ± 0.64 | 88.24 ± 0.51 | **94.06 ± 0.32** | 77.43 ± 1.33 |
| Best CCNN Simplicial | OOM | **89.15 ± 0.32** | 90.32 ± 0.11 | - |
| Best Hypergraph Model | 50.50 ± 0.27 | 81.01 ± 0.24 | 84.52 ± 0.05 | - |

*Table 8.* Results on larger node level datasets, each experiment run with 5 seeds. We report accuracy for Amazon Ratings and Roman Empire, and AUC-ROC for Minesweeper and Questions. The values for the best CCNNs and hypergraph models are extracted from TopoBench (Telyatnikov et al., 2024).

## H.2. More advanced GNNs

We include a subset of experiments with the same protocol as in Table 1 using GATv2 (Brody et al., 2021) and PNA (Corso et al., 2020) in the cellular domain. Results show how the GCCNs built with these models perform consistently well across node-level and graph-level tasks on the cell domain, often $< 1\sigma$ of the best (standard-GNN) GCCN as reported in Table 1, but only outperform them on MUTAG.

*Table 9.* Cross-domain, cross-task, cross-expansion, and cross-$\omega_{\mathcal{N}}$ comparison of GCCN architectures built with GATv2 (Brody et al., 2021) and PNA (Corso et al., 2020) with top-performing GCCNs from Table 1 and benchmarked on TopoBench (Telyatnikov et al., 2024). Best result is in **bold** and results within 1 standard deviation are highlighted blue . Experiments are run with 5 seeds. We report accuracy for classification tasks and MAE for regression.

|  | Graph-Level Tasks | | | | Node-Level Tasks | | |
|---|---|---|---|---|---|---|---|
| Model | MUTAG (↑) | PROTEINS (↑) | NCI1 (↑) | NCI109 (↑) | Cora (↑) | Citeseer (↑) | PubMed (↑) |
| Cellular | | | | | | | |
| CCNN (Best Model on TopoBench) | 80.43 ± 1.78 | 76.13 ± 2.70 | 76.67 ± 1.48 | 75.35 ± 1.50 | 87.44 ± 1.28 | 75.63 ± 1.58 | 88.64 ± 0.36 |
| GCCN $\omega_{\mathcal{N}}$ = Best Standard GNN | **86.38 ± 6.49** | 74.41 ± 1.77 | **78.23 ± 1.47** | 77.10 ± 0.83 | 88.57 ± 0.58 | 75.89 ± 1.84 | 89.40 ± 0.57 |
| GCCN $\omega_{\mathcal{N}}$ = PNA | 83.83 ± 6.31 | 73.91 ± 2.63 | 77.24 ± 1.72 | 76.5 ± 0.88 | 87.27 ± 0.64 | 74.63 ± 1.32 | 86.34 ± 0.38 |
| GCCN $\omega_{\mathcal{N}}$ = GATv2 | **86.38 ± 4.15** | 72.54 ± 3.3 | 77.78 ± 0.94 | 77.04 ± 0.63 | 85.11 ± 0.46 | 72.03 ± 2.54 | 88.32 ± 0.38 |

# I. Performance versus Size Complexity

In, Fig. 10, we extend Fig. 5 to all benchmark datasets. As before, we keep GCCN layers and GNN sublayers in each subplot constant, matching those of the best model of that dataset.

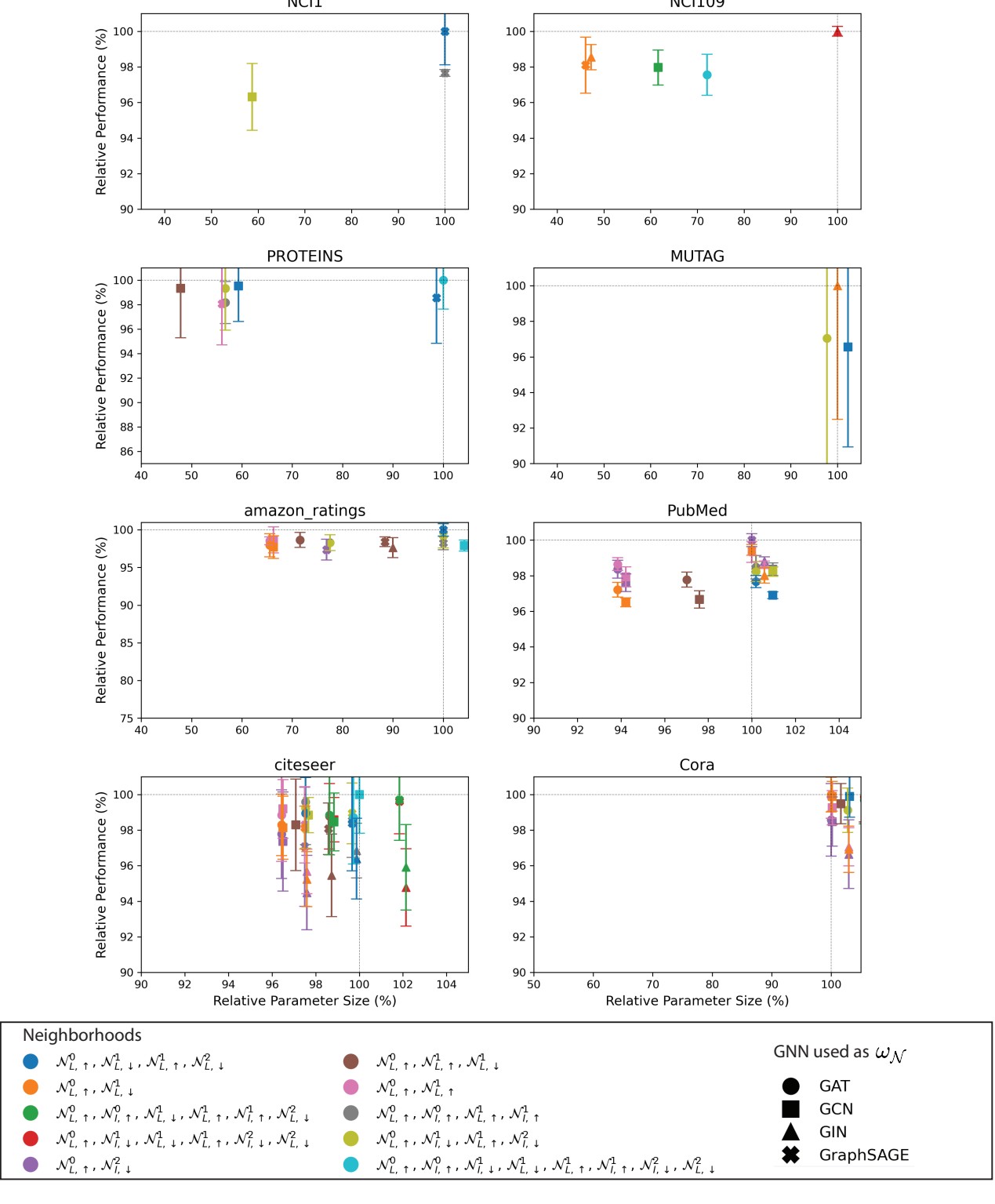

*Figure 10.* Performance versus size, scaled to best model. We consider models within 10% of the best performance on each dataset.

