# OpenReview forum: "TopoTune: A Framework for Generalized Combinatorial Complex Neural Networks"
_ICML.cc/2025/Conference — ICML 2025 poster_

### Official Review · Reviewer_478c · 2025-02-27

**Overall Recommendation:** 3

**Summary:**

The paper suggests a general way to lift GNN architectures to work with simplicial and cell complexes. They implemented their project within a well-known benchmarking suite. The projected is completed with a benchmarking effort covering training on multiple classical graph datasets.

**Claims And Evidence:**

The empirical power of the method is only somewhat supported - the experiments are run on extremely small and outdated datasets and it might be a good idea to mention that pure GNNs do a lot better on those. (For example a simple GIN from 2018 achieves more or less identical performance on MUTAG and newer methods are not even tested on them any more). This may be a problem of the field of topological deep learning that modern benchmarks are lacking.

Further claims (general method etc) are well-supported.

**Essential References Not Discussed:**

-

**Experimental Designs Or Analyses:**

I did not verify the experiments myself as the code is not yet available.

The authors state that they did only use default configurations for all methods and did not perform any hyperparameter tuning. I would like to argue that this is often a very bad idea leading to incomparable numbers between architectures that are too often not representative of the methods' true performance.

**Methods And Evaluation Criteria:**

I would like to argue that the datasets are heavily outdated and should no longer be used for any scientific claims.

Also the base models are a bit simplistic, GATv2 (or better transformerconv) as message passing, as well as architectures such as GatedGCN or PNA typically work better than the basic ones used in the experiments here.

**Other Comments Or Suggestions:**

none

**Other Strengths And Weaknesses:**

Strengths:
- relatively easy to read, also for non-experts in TDL

Weaknesses:
- the general construction that is suggested is not highlighted in terms of any structure, its just part of section 4 and the key observation/construction could have been highlighted a lot more. Also an illustration of how any GCN architecture is turned into a TDL architecture could have been nice.
- the experiments (as mentioned earlier)

**Questions For Authors:**

How did you choose datasets and base models?

Would it be possible to re-run the experiments on e.g. molPCBA and malnet-tiny including automatic hyperparameter tuning?

Is it really common that the whole lifting procedure does not help with the performance? As the absolute values reported in table 1 are more or less what one would expect of a pure GNN on the same datasets (at least if there is still data on those).

**Relation To Broader Scientific Literature:**

looks good to me, but I am also from the GNN side and not the TDL side.

**Theoretical Claims:**

The theoretical claims look ok, but I have not checked the proofs in the appendix. It would have been nice to mention the main proof idea in one sentence in the main paper, indicating whether the proof is straightforward and what the property hinges on.

---

> ### Author Rebuttal · Authors · 2025-03-31
>
> Thank you for your thoughtful comments. We believe they strengthened our work. We are happy to read you found the work accessible for someone not coming from TDL.
>
> **Note on Reviewer's Summary:** GCCNs and TopoTune extend not only GNNs but also non-GNN neural networks. Our framework is also not limited to simplicial and cellular complexes—it generalizes to combinatorial complexes and other higher-order relational structures. While our experiments focus on simplicial and cellular domains, the methodology itself is broadly applicable.
>
> **Responding to performance concerns:** The values in Table 1 serve a benchmarking purpose rather than direct comparison to end-to-end optimized GNNs. TopoBenchmark ensures controlled comparisons by standardizing key components—encoder/decoder design, readout, dataset splits, omission of edge features, and so on. While this impacts absolute performance, it enables a controlled evaluation of architectural differences, where we see GCCNs outperform prior TDL works and standard GNNs when subjected to these constraints.
>
> **Explanation of proofs:** We agree that it is important to briefly describe the idea of the proof. In the updated manuscript, we will add: “Proof of Prop. 4.1 relies on setting the $\omega_\mathcal{N}$ of a GCCN to a simple, single-layer convolution. Proof of Prop 4.2  hinges on the node-wise permutation equivariance of the $\omega_\mathcal{N}$ and the permutation invariance of the inter-neighborhood aggregation. Proof of Prop. 4.3 shows that GCCNs surpass CCNNs in expressivity by relating CCNNs to WL and GCCNs to $k$-WL on augmented Hasse graphs.”
>
> **Weaknesses**
> - Beyond section 4, the general construction (GCCNs) are introduced in the introduction as part of the contributions both conceptually  and empirically. Introducing them before necessary TDL background is tricky–we would appreciate any input on this. A GCCN is pictured in Fig. 1, showing how it is built from $\omega_\mathcal{N}$ (ex.: GCN, see caption line 70).
> - Experiments: We have uploaded an anonymized version of the repository here: https://anonymous.4open.science/r/TopoBench-1F1C/topobench/nn/backbones/combinatorial/gccn.py. Due to the sheer amount of dataset/domains considered, it would be too expensive to run hyperparameter tuning for each of the roughly 50 GCCNs being tested. While we agree tuning would certainly lead to better performance, we argue these results are sufficient to show the superior performance of GCCNs over CCNNs.
>
> **Questions**
> 1. Addressing in two parts:
> - Datasets: The datasets, albeit not recent, were chosen because they are used in TopoBenchmark and thus represent the current norms for benchmarking TDL models. In the future, we will continue testing GCCNs as new datasets inevitably appear in TopoBenchmark. See Q2 for added larger datasets.
> - Models: we chose the base architectures largely based on the vanilla GNNs that are often used as inspiration for TDL models. For example, GAT is the inspiration for CAN (Giusti et al.) and GSAN (Battiloro et al. arxiv.org/abs/2309.02138), GCN for SCNN (Maosheng et al. arxiv.org/abs/2110.02585), and GIN for CWN (Bodnar et al., arxiv.org/abs/2106.12575). We aim to show the TDL community how simple it can be to leverage pre-existing infrastructure from the well-established GNN community. We now also include two more recent GNNs (see Q2).
> 2. We improve the strengths of our empirical results.
> - Dataset-wise, we now include 3 larger node-level benchmark datasets (Amazon Ratings, Roman Empire, Minesweeper) that our machine can support memory-wise and that CCNNs have previously been benchmarked on (due to strict word limit here, table can only be in paper). Summary: GCCNs achieve similar performance to regular CCNNs, outperforming them by a significant margin on Minesweeper. We note that all TDL models are constrained by available liftings, as large graph-based datasets significantly increase in size (see arxiv.org/abs/2409.05211 for active research efforts here).
> - Model-wise, we now include experiments with GATv2 and PNA in the cellular domain. Results (which we will include in the updated paper) show how the GCCNs built with these models perform consistently well across node-level and graph-level tasks on the cell domain, often <1$\sigma$ of best standard-GNN GCCN, but only outperform them on MUTAG.
>
> 3. Lifting generally does improve performance, as seen in comparisons between CCNNs and vanilla GNNs (TopoBenchmark Table 1). We will add a row in our Table 1 showing that standard GNNs match GCCNs/CCNNs in 3 out of 8 datasets, but never outperform them. We also mention very recent work (e.g., Battiloro et al. arxiv.org/abs/2405.15429) showing that simple, general-purpose TDL models outperform GNNs heavily tailored for specific tasks (e.g. molecules).
>
> To conclude, your suggestions helped us summarize theoretical results, contextualize contributions w.r.t. GNNs, and expand our experiments. Please let us know if any concerns remain.

---

> > ### Comment · Reviewer_478c · 2025-04-02
> >
> > I thank the authors for their detailed and helpful rebuttal and also on the note that the proposed method is more generally applicable than I thought. I also highly appreciate the added experiments on more datasets and stronger base models, I think those will clearly strengthen the paper.
> >
> > It seems like the dataset complaint is something that is generally problematic in the whole TDL community (but at least the evaluation procedure and train test splits are fixed which was not always the case when those datasets were still in use in the GNN community). I thank you for also providing results on more relevant graph datasets (which for me makes the results a lot stronger as MUTAG, DD, ENZYMES, ... are considered the MNIST of GNNs that are no longer driving research). I sincerely hope that TopoBench is going to improve their dataset assortment soon as more interesting datasets have been driving research in other fields.
> >
> > This leaves for me mostly the question on hyperparameter tuning and whether a systematic benchmarking approach can exist without it. Personally, I would say that benchmarking is not systematic if hyperparameters are not tuned at all. Since one of the contributions is a systematic benchmarking approach, I feel that this is only partially achieved. For the concrete results, I agree that even without proper tuning, the advantage against other topological models is clear.
> >
> > In terms of practically usable hyperparameter search, often a simple random search will point at good combinations of hyperparameters making sure that no particularly bad hyperparameter configuration was chosen. If inherently supported by some framework, the additional effort may still be manageable.

---

> > > ### Author Response · Authors · 2025-04-04
> > >
> > > We are happy to hear you found our response helpful in better explaining the contribution and the additional experiments to be strong. We sincerely thank you for updating your score to reflect that.
> > >
> > > To address the question of hyperparameter tuning: we understand that some amount of traditional tuning is important to any benchmarking system and appreciate the suggestion. We have performed some targeted hyperparameter tuning for a subset of GCCNs (built from GCN and GIN across all neighborhood structures in Table 1) on four datasets lifted to the simplicial domain. Specifically, we focused on hyperparameters that are independent of the base architecture—encoder dropout, encoder hidden features, and learning rate.
> > >
> > > Our findings indicate that the best hyperparameter combinations yield results that generally remain within one standard deviation of previously reported values (see tables below). This suggests that tuning these parameters at the GCCN level has limited impact, at least for the values we consider. Optimizing base GNN-specific hyperparameters (e.g., hidden dimensions, dropout) may be more influential, or a more refined strategy of search going beyond systematic grid search of a chosen set of values.
> > >
> > > We appreciate the reviewer’s suggestion regarding structured hyperparameter search and its relevance for a practitioner wishing to. apply TopoTune to a real-world scenario. If the paper is accepted, we will explicitly discuss this point and promising directions (ex: GNN level versus GCCN level) for performance gains.
> > >
> > > Please let us know if you have any further questions or concerns.
> > >
> > > --------
> > >
> > > **Table: Hyperparameter search results**
> > >
> > > | Model                           |                          | MUTAG ($\uparrow$) | PROTEINS ($\uparrow$) | NCI1 ($\uparrow$) | NCI109 ($\uparrow$) |
> > > |---------------------------------|--------------------------|--------------------|-----------------------|-------------------|---------------------|
> > > | Simplicial                      |                          |                    |                       |                   |                     |
> > > | GCCN $\omega_\mathcal{N}$ = GCN | from Table 1             | 74.04 ± 8.30       | 74.91 ± 2.51          | 74.20 ± 2.17      | 75.76 ± 1.28        |
> > > |                                 | best from hyperp. search | 74.29 ± 4.2        | 75.15 ± 2.32          | 73.54 ± 0.14      | 73.04 ± 1.52        |
> > > | GCCN $\omega_\mathcal{N}$ = GIN | from Table 1             | 85.96 ± 4.66       | 72.83 ± 2.72          | 76.67 ± 1.62      | 75.64 ± 1.94        |
> > > |                                 | best from hyperp. search | 83.5 ± 4.51        | 73.56 ± 2.91          | 76.19 ± 1.14      | 75.87 ± 1.62        |

---

### Official Review · Reviewer_DsXg · 2025-03-10

**Overall Recommendation:** 3

**Summary:**

This paper aims to further topological deep learning by allowing for the easy adaption of any GNN into network for cell complexes. The basis for their method is representing the cell complexes with augmented hesse graphs, running GNNs on these graphs separately and then combining features from each of the graphs

They show that their class of networks has the same expressive power as CCNN, unlike other graph based methods, but allow for easy plug-and-play GNN backbones (on the Hesse graphs). This allows for users to more easily access TDL methods without the loss of expressivity. Although this idea is fairly simple, it has the potential to make a significant impact.

**Claims And Evidence:**

Yes

**Essential References Not Discussed:**

None that I am aware of

**Experimental Designs Or Analyses:**

I did not check this

**Methods And Evaluation Criteria:**

Yes

**Other Comments Or Suggestions:**

N/A

**Other Strengths And Weaknesses:**

N/A

**Questions For Authors:**

Can GNNs that don't fit the strict definition of message passing, e.g. ChebNet be incorporated into your software?

**Relation To Broader Scientific Literature:**

This should help accelerate research into TDL

**Theoretical Claims:**

I skimmed but did not thoroughly check the proofs. I view the theory as fairly "standard"/"unsurpising", but I am reasonably confident it is likely correct (or at least not substantially wrong)

---

> ### Author Rebuttal · Authors · 2025-03-31
>
> Thank you for your review. We are happy to read that you believe there is potential for significant impact and acceleration of TDL.
>
> We answer your main question here. Yes, any neural network (not even necessarily a GNN) can be easily incorporated into TopoTune. Practically speaking, as long as the neural network can be imported as a PyTorch module, it can be selected as the chosen base architecture ($\omega_\mathcal{N}$) when defining the GCCN.
> As such, selecting a model like ChebNet would simply mean choosing a spectral graph network as the $\omega_\mathcal{N}$ function. In this case, features inside each neighborhood would first be spectrally updated (step B of Fig. 1) and then neighborhood-level features would be spatially aggregated (step C of Fig. 1).
>
> We will better emphasize this fact in the paper at line 324 col 1: “Differently from the work in Hajij et al., 2023, the fact that GCCNs can have arbitrary neighborhood message functions implies that non message-passing TDL models can be readily defined. For example, one could choose $\omega_\mathcal{N}$ to be a spectral graph neural network such as Defferrard et al., 2016.”
>
> Please let us know if you have any other questions.

---

### Official Review · Reviewer_7uco · 2025-03-13

**Overall Recommendation:** 3

**Summary:**

The paper introduces Generalized Combinatorial Complex Neural Networks (GCCNs) extending Topological Deep Learning (TDL) models to the combinatorial domain. It generalizes Combinatorial Complex Neural Networks (CCNNs), offering improved expressivity and performance, often with reduced model complexity. To facilitate the design and training of these TDL models, they present TopoTune, a lightweight software framework that simplifies the creation of TDL architectures.

**Claims And Evidence:**

Yes

**Essential References Not Discussed:**

N/A

**Experimental Designs Or Analyses:**

Yes

**Methods And Evaluation Criteria:**

Yes

**Other Comments Or Suggestions:**

See weaknesses and questions.

**Other Strengths And Weaknesses:**

- The paper achieves multi-level information aggregation through an ensemble of augmented hasse graphs. Could a similar effect be obtained using combinatorial or Hodge Laplacians, as explored in https://arxiv.org/pdf/2403.06687 and https://arxiv.org/abs/2309.12971, which aim to integrate information across different ranked simplices via spectral filtering?

- While the proposed Generalized Combinatorial Complex Network (GCCN) is theoretically broad, the experiments are limited to simplicial and cellular complexes—potentially due to memory constraints or challenges in the lifting procedure. Given this, would it be useful to compare GCCNs with models for cell and simplicial complexes, such as MPSN or CWN, to evaluate the method’s efficacy across different topological representations?

- The paper claims to address Open Problem 1 from the TDL position paper. However, the experiments primarily focus on simple graph and node classification tasks, lacking diversity in application domains. This is also valid for open problem 3 as the paper proposes a method not a benchmarking software like TopoBenchmark. The authors should be cautious in making such claims, as the current scope of evaluation may not fully substantiate solving the broader open problem, potentially leading to misleading interpretations.

**Questions For Authors:**

- Are the hyper-parameters of baselines also optimized as they are done for the GCCN?
- How does the transformer version of GCCN compare to this https://arxiv.org/pdf/2405.14094? It seems it has already solved the open problem 11, as mentioned in the TDL position paper.

**Relation To Broader Scientific Literature:**

- This work builds on prior Topological Deep Learning (TDL) work by generalizing Combinatorial Complex Neural Networks (CCNNs), which are more expressive than GNNs.
- The authors prove their method, i.e., Generalized CCNNs (GCCNs) subsume CCNNs, achieving comparable or better performance with lower model complexity.
- The paper introduces TopoTune, a software framework that simplifies the design and training of TDL models, similar to how PyG and DGL standardized GNNs.

**Theoretical Claims:**

No

---

> ### Author Rebuttal · Authors · 2025-03-31
>
> Thank you for your helpful feedback. We address comments and questions below.
>
> **(Weakness 1) Hodge Theory and Spectral Filtering:** Thank you for your thoughtful comment. While the cited works integrate information across simplices, they are restricted to simplicial complexes due to their reliance on cohomology for defining Hodge Laplacians (not possible for hypergraphs) and the PSD property of Flower-Petals Laplacians (not possible for cell complexes or hypergraphs). Our approach, in contrast, generalizes to diverse higher-order domains without requiring specific spectral notions (e.g., no cohomology theory nor Flower-Petals Laplacian have been developed yet for general combinatorial complexes). However, any spectral model—whether graph-, simplicial-, cellular-, or hypergraph-based—can be used as $\omega_\mathcal{N}$​ (step B, Fig. 1) and combined with spatial aggregation (step C). This highlights the flexibility of GCCNs and TopoTune in enabling new topological architectures. We will clarify that spectral networks can serve as base architectures for non-message-passing models.
>
> **(Weakness 2) Experiments in simplicial and cellular domains:** You are correct that we limit our experiments to these two domains in part because of computational cost and in part because of the lack of available “standard” liftings for hypergraphs and combinatorial complexes. However, we do in fact compare GCCNs to existing models by comparing the models to the “Best available model in TopoBenchmark”. These models include CWN and MPSN.  In the revised version, we will specify which model is the best available model in TopoBenchmark, and provide a complete list of all models included in TopoBenchmark (and thus that we compare against).  We emphasize that the purpose of Table 1 is exactly to evaluate the method’s efficacy across different topological domains, with different choices of base architecture ($\omega_\mathcal{N}$).
>
> **(Weakness 3) Open problems:**
> - (OP 1) While we agree the datasets used in these experiments are nothing new, we argue that the unprecedented ease TopoTune provides in defining and training new architectures makes TDL a much more practical option for real-world applications, especially those for which custom GNNs have already been developed.  In the updated manuscript, we will rephrase the claim to better emphasize this (line 106, col 2): “Using TopoTune, practitioners can, for the first time, easily define and iterate upon TDL models, making TDL a much more practical tool for real-world datasets (OP 1: need for accessible TDL).”
> - (OP 3: need for standardized benchmarking) Our work provides a standardized benchmarking of GCCNs. Unlike traditional studies that tune a single model across datasets, we train ~50 models (varying base architectures and neighborhoods) and compare their performance across multiple datasets and topological domains. This systematic approach is a notable contribution, addressing the field’s reliance on comparisons under heterogeneous training conditions and marking a first step toward solving OP 3. We will clarify this in the contributions: “Unlike prior works that compare models under heterogeneous conditions, our systematic benchmarking provides a controlled evaluation of GCCNs across diverse architectures, datasets, and topological domains.”
>
> **Questions**
> 1. Beyond the sweep of possible base architectures and neighborhoods, we did not do a traditional “optimization” of hyperparameters for the GCCNs in the sense that we only considered one set of training hyperparameters for each combination of GCCN and task. This set of hyperparameters was selected from the defaults proposed by TopoBenchmark. If there was no default available, we picked the lowest value considered in TopoBenchmark reported grid search. To answer your question, the CCNNs in TopoBenchmark's hyperparameters were obtained with a wide grid search, as described in Appendix C.2 of their work (https://arxiv.org/pdf/2406.06642). We will clarify this in the “Experimental Setup” subsection: “While CCNN results reflect extensive hyperparameter tuning by Telyatnikov et. al, 2024 (see that work's Appendix C.2 for details),...”
>
> 2. When we mention OP 11, we are specifically addressing the need for cross-domain attentional TDL. Just like in other areas of TDL, existing attentional works (including Ballester et al) are limited to one domain (in this case, cellular complexes). Because of its integration into TopoTune, the GCCN architecture built with a Transformer base architecture is inherently able to accommodate any topological domain.
>
> Thank you again for your time and review. Your comments helped us clarify our discussion on spectral methods and better articulate the scope of experiments. We have also improved the clarity and nature of our contributions. Please let us know if you have further questions.

---

### Official Review · Reviewer_hN8B · 2025-03-16

**Overall Recommendation:** 4

**Summary:**

This paper introduces generalized combinatorial complex neural networks (GCCNs), which provide a general technique for turning any existing (graph) neural network architecture into a topological network, which operates on combinatorial complexes. Their method operates by turning a combinatorial complex into a series of graphs, where each one is defined by a neighborhood function of the combinatorial complex. The proposed GCCN is proven to be more expressive than the existing CCNNs, preserves the appropriate permutation symmetries, and also tends to outperform CCNNs on TopoBenchmark graph tasks. In computational complexity, it interpolates between GNNs and CCNNs. The paper also presents a codebase, TopoTune, for implementing GCCNs.

**Claims And Evidence:**

Yes

**Essential References Not Discussed:**

n/a

**Experimental Designs Or Analyses:**

No issues found

**Methods And Evaluation Criteria:**

The benchmark dataset makes sense, as well as the comparison to CCNNs. However, it would have also been helpful to compare to non-CCNN baselines, as in the original TopoBenchmark paper.

**Other Comments Or Suggestions:**

As someone who is not very embedded in the TDL community, it would expand the reach of your paper to better motivate (briefly) the field of TDL in the intro. The first paragraph does a nice job of this, conceptually; are there empirical works that you can cite, which demonstrate the benefits of capturing these multi-way relationships? For example, what are some datasets on which TDL is SOTA?

**Other Strengths And Weaknesses:**

Strengths: The paper is clearly written and outlines their contributions clearly, which is a general architectural framework that seems to address open challenges in TDL. Moreover, the paper produces a library, TopoTune, which promises to be a useful mechanism for fast prototyping of TDL architectures on TopoBenchmark and for reproducing their results with GCCNs. Armed with TopoTune, the experimental results cover a wide range of GCCN architectures, which often outperform existing CCNNs. Although I am not very familiar with TDL, these seems like a valuable contribution to the field.

Weaknesses: The paper only compares GCCNs to CCNNs, rather than other non-topological, perhaps more standard architectures (vanilla GNNs, transformers, etc). Also, their GCCN can be instantiated with many different choices of base graph architecture, but it is not clear how to choose one in practice.

**Questions For Authors:**

1. Is there a reason for only comparing with CCNNs on the tasks in TopoBenchmark?
2. Relatedly, the best MAE reported for ZINC in Table 1 is 0.19. paperswithcode reports the SOTA number as 0.056, as achieved by chromatic self-attention (Menegaux et al 2023), a non-topological or combinatorial architecture. Does GCCN achieve SOTA on any of the datasets in TopoBenchmark, when considering architectures other than CCNNs? If not, how do the authors view the utility of the benchmark, and their GCCN architecture, in a broader context?
3. Based on Table 1, the specific architecture choice of $\omega_{\mathcal{N}}$ that performs best varies by task. In practice, how then should one choose $\omega_{\mathcal{N}}$? Is there any way to avoid exhaustive search and retraining, e.g. by using task-specific insights?

**Relation To Broader Scientific Literature:**

This paper builds naturally on Hajij et all, which introduced combinatorial complexes and architectures for them (CCNNs). The new class of architectures, GCCNs, extends CCNNs. It also answers open challenges in TDL as posed by Papamarkou et al 2024. It uses, as a base architecture for the message passing on each created augmented Hasse graph, methods like GIN, GCN, GraphSAGE, GAT, etc, thereby building on those papers.

**Theoretical Claims:**

I did not check correctness of proofs

---

> ### Author Rebuttal · Authors · 2025-03-31
>
> Thank you for your time as well as your positive and thoughtful review. We are happy to read that the method made sense and the contribution was well justified. We address the raised points about weaknesses and questions below.
>
> **(Weaknesses) Comparing to standard architectures:** We initially only compared GCCN on topological domains for fairness reasons (comparing topological domains amongst themselves), which is why graphs were not included. However, in practice, we completely agree it is very helpful to know if a simple GNN does better. In the updated manuscript, we will add a line to Table 1 that shows the best-performing GNN tested in TopoBenchmark on each dataset. We can then see that standard GNNs achieve comparable results only in 2 of the 16 dataset/domain combinations GCCNs were tested on (PROTEINS in the cellular domain, PubMed in the simplicial domain). This will now be specified in the Results subsection “GCCNs outperform CCNNs.”
>
> **(Weaknesses, Q3) Choosing an optimal base architecture:** You are correct that we do observe significant variation in performance between GNNs (see section “Impactfulness of GNN choice is dataset-specific.”, line 411 col 2, as well as Fig. 5). By better leveraging GNN works in TDL, practitioners can build off the extensive benchmarking work performed in the GNN field (see for example the benchmark study https://arxiv.org/abs/2003.00982). For example, it comes as no surprise that the base architecture of GIN is a good choice for the ZINC dataset, as that is how it appears in the GNN world as well. We leave the study of further pre-training optimization of hyperparams such as choice of topological domain and choice of neighborhoods to future work. However, we emphasize that one of the goals of a principled framework like TopoTune is to make such a study, that would be otherwise extremely hard, feasible.
>
> **(Comments) Better motivating TDL:** TDL architectures perform well in general but are particularly valuable when higher-order multiway interactions matter significantly. Examples include citation networks (Battiloro et al., https://arxiv.org/abs/2309.02138) and human skeleton-based action recognition (Hao et al., https://ieeexplore.ieee.org/stamp/stamp.jsp?tp=&arnumber=9329123). A notable SOTA TDL model is TopoRouteNet, used in computer network modeling (Bernardez et al., https://arxiv.org/html/2503.16746v1). Recent work (Battiloro et al., https://arxiv.org/abs/2405.15429) also highlights TDL’s potential in molecular tasks, demonstrating that simple, general-purpose TDL architectures can outperform models heavily tailored for molecules. That said, benchmarking our methods on similar datasets to prior works is a crucial step toward broader application-specific adoption. We will add a note on successful TDL applications at the end of the “TDL Research Trend” section of the introduction.
>
> **Choosing tasks (Q1).** TopoBenchmark consolidates the most commonly used benchmark tasks in TDL, as well as the highest-performing, most well known models in TDL. By including this variety of tasks, GCCNs benefit from a comprehensive test against the field. Going beyond that, we also specifically use the TopoBenchmark platform to ensure fair comparison amongst architectures, as all the tasks are homogenized in, for example, data split, reported performance metric, and so on (see point below).
>
> **SOTA Results (Q2).** This performance gap on ZINC primarily arises from the benchmarking framework used in our work. As TopoBenchmark (Telyatnikov et al.) standardizes several components—such as encoder/decoder, readout, dataset splits and the omission of edge features—it ensures a more controlled comparison of model backbones but may come at the cost of absolute SOTA performance. Generally, high-performing GNN models, including Chromatic Self-Attention (Menegaux et al. 2023), are designed as end-to-end architectures with carefully tuned encoders and readouts, whereas TopoBenchmark enforces uniformity in these components to isolate and assess the impact of architectural differences.
>
> In this context, while GCCN may not achieve SOTA performance, the benchmark enables meaningful insights into the role of topological architectures by minimizing confounding factors. This approach aligns with our broader goal: rather than optimizing for peak performance on individual datasets, we aim to establish a fair and standardized evaluation framework that can help guide future architectural improvements. Importantly, TopoBenchmark is flexible enough to accommodate advances in GNN design. This could look like building a GCCN with Chromatic Self-Attention as a base architecture ($\omega_\mathcal{N}$).
>
> **Conclusion.** Thank you again for your review. Your feedback helped us make (key) comparisons to GNNs, better motivate TDL, and improve our contextualization of results. We appreciate your insights and believe they have made the manuscript stronger.

---

### Decision · Program_Chairs · 2025-05-01

**Decision:**

Accept (poster)

**Comment:**

The authors present TopoTune, a framework that generalizes Graph Neural Networks to Topological Neural Networks by introducing Generalized Combinatorial Complex Neural Networks (GCCNs). All reviewers gave a positive rating. It is the AC’s opinion that the work makes a nice contribution to Topological Deep Learning by providing a standardized, accessible framework. The software implementation further enhances the paper's impact by democratizing TDL research.  I recommend accepting this paper.